



# It could have been much worse: spatial counterfactuals of the July 2021 flood in the Ahr valley, Germany

Sergiy Vorogushyn[1], Li Han[1], Heiko Apel[1], Viet Dung Nguyen[1], Björn Guse[1,2], Xiaoxiang Guan[1], Oldrich
Rakovec[3,4], Husain Najafi[3], Luis Samaniego[3,5], Bruno Merz[1,5]

[1] Section Hydrology, GFZ German Research Centre for Geosciences, 14473 Potsdam, Germany

[2] Department Hydrology and Water Resources Management, Kiel University, 24098 Kiel, Germany

[3] Department Computational Hydrosystems, Helmholtz Centre for Environmental Research – UFZ, 04318
Leipzig, Germany

[4] Faculty of Environmental Sciences, Czech University of Life Sciences, 16500 Prague, Czech Republic

[5] Institute for Environmental Sciences and Geography, University of Potsdam, 14476 Potsdam, Germany

*Correspondence to*: Sergiy Vorogushyn (sergiy.vorogushyn@gfz-potsdam.de)

**Abstract.** After a flood disaster, the question often arises: "What could have happened if the event had gone differently?" For example, what would be the effects of a flood if the path of a pressure system and the precipitation field had taken a different trajectory? In this paper, we use alternative scenarios of precipitation footprints shifted in space, the so-called "spatial counterfactuals" to generate plausible but unprecedented events. We explore the spatial counterfactuals of the deadly July 2021 flood in the Ahr Valley, Germany. We drive a hydrological model of the Ahr catchment with precipitation fields of this event systematically shifted in space. The resulting discharge is used as a boundary condition for a high-resolution two-dimensional hydrodynamic model. We simulate changes in peak flows, hydrograph volumes, maximum inundation extent and depths and affected assets and compare them to the simulations of the actual event. We show that even a slight shift of the precipitation field by 15-25 km eastwards, which does not seem implausible due to orographic conditions, causes an increase in peak flows at the gauge Altenahr of about 32 % and of up to 160 % at the individual tributaries. Also, significantly larger flood volumes of more than 25 % can be expected due to this precipitation shift. This results in significantly larger inundation extents and maximum depths at a number of analyzed focus areas. For example, in the focus area around Altenahr, the increase of mean and maximum depth of up to 1.25 m and 1.75 m, respectively, is simulated. The presented results should encourage flood risk managers as well as the general public to meet precautionary measures for extreme and unprecedented events.

## 1. Introduction

On 14-15 July 2021, an exceptional flood event struck a vast region in western Germany, Belgium, Luxemburg and the Netherlands, causing more than 230 deaths and total economic loss of up to 50 billion euro (Szöni et al., 2022). In Germany alone, more than 180 people lost their lives and loss estimates range between 35 and 40 billion euro. The Ahr river valley in the Eifel mountains was a hotspot of flood impact with 134 deaths and two



people still missing (DKKV, 2022). This is the highest human life loss due to a flood catastrophe in Germany since the storm surge in 1962 in Northern Germany.

In the first half of July 2021, a low-pressure system formed over the Northern German Plains resulting in a southwestward air flow strongly enriched with moisture from the Baltic Sea and the North Sea (Mohr et al., 2023). As a result, precipitation sums of up to 150 mm were recorded within 15 to 18 h in parts of the Ahr catchment. Using the hourly radar-based RADOLAN data (Weigl and Winterrath, 2009, Winterrath et al., 2018) and the daily station-based HYRAS dataset (Rauthe et al., 2013), Mohr et al. (2023) estimated the return period of precipitation in the order of 500 years in parts of the Ahr catchment. Kreienkamp et al. (2021) and Tradowsky et al. (2023) estimated a return period of about 400 years for daily precipitation sums in the period from April to September by pooling the HYRAS data over a larger region between the North Sea and the Alps. This heavy and intense rainfall resulted in an extreme catchment response, with strong field evidence for massive overland flow, where ephemeral drainages in forests and grassland turned into concentrated streams (Dietze et al., 2022). The rapid water rise within a few hours led to unprecedented water levels of up to 10 m exceeding the instrumental record since 1947 and the historical water marks since 1804. The gauges Altenahr and Müsch (Figure 1) were destroyed, so no discharge could be recorded instrumentally. The post-event peak flow reconstructions at gauge Altenahr range from 750 to 1100 $m^3 s^{-1}$. Roggenkamp and Herget (2022) suggested a peak flow of 1000 $m^3 s^{-1}$ and higher based on the recorded wrack marks, surveyed topography and the application of Manning's equation with typical roughness values. The model-based flow reconstruction by Berker et al. (2022) resulted in somewhat lower values between 750 and 1000 $m^3 s^{-1}$, partly considering the backwater effect due to clogging of several bridges. The reconstructed peak flow exceeded the highest instrumental record of 236 $m^3 s^{-1}$ in 2016 by three to more than four times. It reached about the same level as the reconstructed peak flow of the 1804 summer flood, for which a few historical high-water marks are available (Roggenkamp and Herget, 2014). Remarkably, the water levels during the 2021 flood exceeded those of 1804 by more than 2 m (Mohr et al., 2023), probably due to the aforementioned bridge clogging and resulting backwater effects, but also due to denser settlements and higher macroscopic roughness at present time. Considering the historical floods of 1804, 1888, 1910, 1918, 1920 and recent instrumental records, Vorogushyn et al. (2002) estimated the local return period at gauge Altenahr of more than 8500 years based on the peak flow reconstruction by Roggenkamp and Herget (2022). Due to limited records, the very high skewness of the timeseries and the poor fit of the statistical model to the extremes, the return period estimates are associated with very high uncertainties.

The likelihood and intensity of extreme floods as in July 2021 can increase in a warmer climate due to increased heavy precipitation. Deploying ensembles of regional and global climate models, the extreme event attribution studies of Kreienkamp et al. (2021) and Tradowsky et al. (2023) suggested an increased likelihood of the observed maximum 1-day precipitation to occur in the present climate compared to the pre-industrial state (1.2 °C cooler) by a factor of 1.2 – 9. A further increase in the likelihood by a factor of 1.2 – 1.4 is suggested in a 2 °C warmer climate compared to the pre-industrial state. The maximum 1-day precipitation intensity was suggested to increase by 3.8 – 25%. The estimation by Ludwig et al. (2023) with an increase of 11 – 18% in event precipitation totals in the region around the Ahr catchment for the +2 °C climate falls within the above range estimated by Kreienkamp et al. (2021) and Tradowsky et al. (2023). Ludwig et al. (2023) used the pseudo global warming approach (Schär et al., 1996), in which the temperature changes corresponding to the fixed warming level of +2 °C are prescribed at the initial and lateral boundary conditions of a regional climate model.



They further analyzed the hydrologic response of the Ahr catchment to higher precipitation in a +2 °C warmer climate with a distributed hydrological model. The projected increase in peak flows of up to 39% at gauge Altenahr is alarming and underlines the non-linearity of the hydrologic catchment response.

The severity of the July 2021 flood disaster, but also the adverse potential future changes, call for a set of actions
to improve flood risk management and climate adaptation in the catchment communities. Besides the reassessment of flood design values used for flood hazard mapping and infrastructure planning, i.e., in the range of 30 – 200 y return periods, it is highly valuable to explore extreme and unprecedented scenarios that have not been observed in the past but may occur in the near future (Montanari et al., 2023, in review). Kreibich et al. (2022) concluded from a study of 45 paired subsequent extreme events (floods and droughts) that risk
management in general reduces the impacts of a second event in the same area. Societies however face difficulties in reducing the impacts of unprecedented events if the magnitude of the second event exceeds past experience. The water-proof design for all possible unprecedented scenarios is not possible and too costly. However, some actions can be taken with small additional effort and cost that unfold pivotal effects when unprecedented scenarios are put on the mind map of decision makers. Flood-prone people and crisis managers
need to be prepared for such situations to reduce at least the most harmful consequences such as death toll. Critical infrastructure, e.g., local crises centers, need to be located outside of the potentially affected areas to ensure their functionality during catastrophic situations.

There is a plethora of approaches to constructing scenarios of exceptional events (Merz et al., 2021). A standard approach relies on the extrapolation using extreme value statistics and estimating high return period floods,
which may or may not have occurred in the past in the specific catchments. For example, Apel et al. (2004) upscaled the averaged observed hydrographs from the past floods to the peak flows extrapolated up to 10,000 years from extreme value statistics. These scenarios were used to estimate flood risk along the Rhine River in Germany. In the Ahr valley, Vorogushyn et al. (2022) estimated the 1,000-year flood and analyzed the associated inundation based on extrapolating the GEV distribution considering historical floods. Extrapolations
based on extreme value statistics suffer well-known limitations rooted in the limited sample size, selection of a statistical model and parameter estimation procedure (e.g. Hu et al., 2020). Furthermore, extreme floods are often different from small floods in terms of the atmospheric, runoff generation and river network processes, so the extrapolation may not be valid, see Merz et al. (2022) for discussion. To partly overcome this limitation, stochastic weather generators can be used in combination with hydrological models to continuously simulate
long-term series of events which include unprecedented events (e.g., Falter et al., 2015, Viviroli et al., 2022).

Another set of approaches includes the estimation of probable maximum precipitation (PMP) and associated probable maximum floods (PMF). The World Meteorological Organization (WMO) provides guidelines for estimating PMP containing several methods (WMO, 2009). PMP is typically estimated for storms of various durations for a specific catchment by applying theoretically grounded maximizations to the storm parameters.
Various approaches can be used for the spatial and temporal representation of the PMP in a specific catchment and for the computation of the resulting PMF (Felder and Weingartner, 2017). Recently, approaches have been developed to adjust PMP estimates for non-stationary climate based on information from physically-based climate models (Chen et al., 2017, Visser et al., 2022).


The climate community proposed the future weather or storyline approach, in particular to explore the evolution of extreme weather events and their impacts under future climate conditions (Hazeleger et al., 2015, Shepherd, 2018). To this end, the past synoptic-scale extremes are imposed onto perturbed boundary conditions in climate models, e.g., changed atmospheric composition, land use and/or sea surface temperature. Following this approach, Manola et al. (2018) imposed a heavy summer precipitation event in July 2014 over The Netherlands onto the present and future climate conditions in a high-resolution convection-permitting numerical weather prediction model. The precipitation event generated in this way for future climate conditions unfolds a displaced pattern with an increase in precipitable water per degree of warming of nearly double the Clausius-Clapeyron rate. The above-mentioned pseudo global warming simulations by Ludwig et al. (2023) can also be regarded as a type of future weather or storyline approach.

A fourth approach for developing extreme scenarios is the construction of so-called "perfect storms". The term "perfect storm" denotes an unfavorable superposition of several factors or phenomena that lead to an unprecedented event, whereas these phenomena have previously occurred in isolation (Paté-Cornell, 2012). The term refers to a severe storm that occurred over the North Atlantic in October 1991 as a conjunction of a storm over the US, a cold front from the north and a tropical storm from the south (Paté-Cornell, 2012). In hydrology, an example of a "perfect storm" would be a scenario with an unfavorable superposition of extreme antecedent catchment conditions and extreme precipitation that occurred in isolation, but not in combination. To the best of our knowledge, there is only one study in the hydrological literature that recombined historical snowpack with design precipitation events in Sweden for estimation of design floods for dams and spillways (Bergström et al., 1992).

Finally, past events can be explored by analyzing event properties and processes that could have been worse. This approach, introduced to natural hazards by Woo (2019), provides so-called downward counterfactual scenarios. Downward counterfactuals contrast with upward counterfactuals where things turn for the better. In general terms, counterfactual refers to a possible realization of a past event, upward or downward. Spatial counterfactuals can be defined as a special case of counterfactuals, where the past events are shifted in space. Spatial counterfactuals are an intuitive approach to explore the possibility of unseen and exceptional events in a specific area. Merz et al. (2024) pioneered this concept for flood hazard assessment and explored alternative scenarios of the 10 most damaging floods in Germany. By shifting precipitation fields by several tens of kilometers in space, they explored changes in return periods of peak flows generated by a hydrological model. In a similar vein, Voit and Heistermann (2024, in review) generated spatial counterfactuals from the 10 most severe high precipitation events in 2021-2022 and combined them with more than 22,000 sub-catchments in Germany. The analysis of more than 220,000 combinations resulted in many unprecedented floods across Germany. The results, however, are based on the strong assumption that any of the past high precipitation events could occur anywhere in Germany.

Flood hazard and risk assessment has tremendously advanced in the past decades. The European Union Member States have implemented nation-wide flood hazard mapping, as enforced by the EU Flood Directive (EU, 2007). In Germany, inundation hazard is mapped for a low return period flood (~10 – 20 yr), a high return period event (100 yr) and an extreme scenario (200 – 1000 yr) with some variation between the federal states (Vorogushyn et al., 2022). Similar return periods of 30, 100 and 300 years have been used in Austria (Blöschl et al., 2023). However, exceptional floods exceeding even the mapped extreme scenario continue to occur, such as the July



2021 flood in the Ahr valley and as the 2002 and 2013 floods in the Elbe basin (Schröter et al., 2015). This is to
155 be expected due to the stochastic nature of the flood generation processes within the space of possible event
realizations. In addition, climate change may contribute to the occurrence of exceptional or even unprecedented
events (Robinson et al., 2021). Further, the poor estimation of flood quantiles (Vorogushyn et al., 2022) and the
ignorance of the variety of possible unprecedented events by risk managers, decision makers and the public may
catch everyone by surprise and result in devastating consequences (Merz et al., 2015, Woo, 2019). Hence, a
160 systematic procedure is needed for exploring the space of potential unprecedented events that may turn into
catastrophes (Woo, 2019).

In the present study, we address the challenge of exploring the space of unprecedented flood events in the Ahr
catchment by developing spatial counterfactuals for the July 2021 flood. In particular, we search for downward
counterfactuals by answering the key questions of where, how much and why the intensity and impacts turn out
to be more severe that during the actual flood event. Spatial counterfactuals are constructed by shifting the
observed spatio-temporal precipitation footprint in space. We go beyond the previous studies by Merz et al.
(2024) and Voit and Heistermann (2024, in review) and deploy for the first time a flood process model chain
encompassing hydrology, flood inundation and impact quantification for the analysis of spatial counterfactual
scenarios.

## 2. Study area

We analyze spatial counterfactuals for the Ahr river catchment in western Germany. The catchment, with an area
of about 900 km$^2$, drains the Ahr-Eifel mountains in the German federal states of Rhineland-Palatinate and
Northern Rhine-Westphalia to the Rhine River (Fig. 1). The 86 km long river springs up at the elevation of about
175 520 m a.s.l. and crosses the deeply incised valley down to the Rhine mouth near Sinzig. Several major tributaries
with respective gauges, such as Adenauerbach (gauge Niederadenau), Staffelerbach (gauge Denn), Sahrbach
(gauge Kreuzberg) enter the main river, which is gauged at Müsch, Altenahr and Bad Bodendorf from upstream
to downstream (Fig. 1). The catchment is characterized by shallow soils of primarily clay slate. Large parts of
the catchment are covered by forests, and some grassland at the elevated plateaus. Arable land is particularly
concentrated in the northeastern lowlands, whereas steep slopes in the middle reaches are used as vineyards that
are located between the riverine villages and the mountain ranges (LfU, 2005). The mean annual catchment
precipitation ranges between 550 and 900 mm (HAD, 2024), with mean monthly July totals of 70 mm in 1991-
2020 (Berkler et al., 2022). The mean flow at the gauges Müsch and Altenahr amounts to about 3 and 8 m$^3$ s$^{-1}$
with mean annual flood peak flow of 65 and 90 m$^3$ s$^{-1}$, respectively.

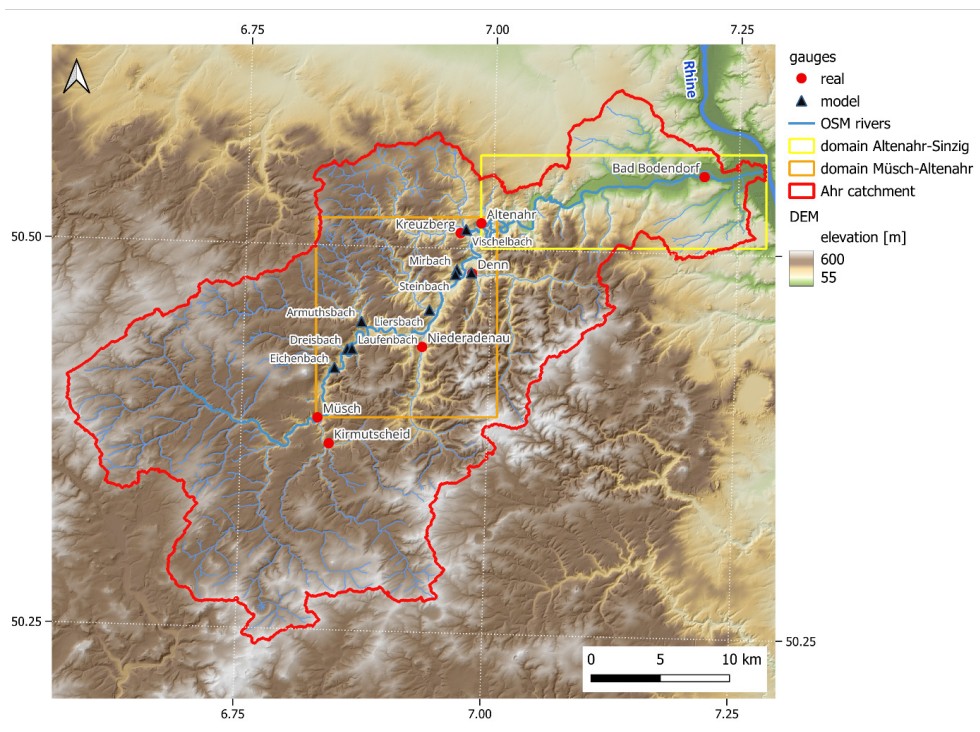

**Figure 1.** Ahr catchment, major river network from the Open Street Map (OSM), Digital Elevation Model
(DEM5), location of the real and virtual ("model") gauges. Virtual gauges represent the locations, where inflow
boundary conditions from mHM to RIM2D model are specified. RIM2D modelling domains Müsch-Altenahr
and Altenahr-Sinzig are shown in orange and yellow boxes, respectively. © OpenStreetMap contributors 2024.
Distributed under the Open Data Commons Open Database License (ODbL) v1.0.

## 3. Data and Methods

### 3.1 Counterfactual precipitation fields

For the development of spatial counterfactuals, we use the E-OBS dataset v.25e of daily precipitation sums at the
resolution of 0.11° x 0.11° (Cornes et al., 2018), which extends till 31 December 2021 and includes the 2021
flood. For the purpose of hydrological modelling, the precipitation fields are re-gridded to the 0.0625° x 0.0625°
grid using bi-linear interpolation. Furthermore, daily precipitation sums are disaggregated to hourly values using
the method of fragments (Guan et al, 2023). A vector of hourly fragments, representing the relative distribution
of hourly precipitation to the daily sum, is obtained from RADOLAN hourly observations. The RANDOLAN
dataset (Weigl and Winterrath, 2009, Winterrath et al., 2018) provides historical, hourly, German-wide, gridded,
highly resolved precipitation data from the combination of the hourly values measured at climate stations with
the precipitation recording of 17 weather radars. The RADOLAN data has a spatial resolution of 1 km and

covers the period from 1 June 2005 to the present. We use the coarser E-OBS data instead of directly applying

RADOLAN hourly fields because of two reasons. First, E-OBS contains consistent precipitation and temperature
data needed for hydrological modelling. Second, the hydrological model used in this study is calibrated with the
E-OBS data as a part of the setup covering German catchments including parts outside of Germany, for which
RADOLAN is not available.

The major trajectory of the atmospheric moisture transport into the affected region was from northeast to

210 southwest on the northern flank of a low-pressure system (Mohr et al., 2023). Hourly precipitation footprints
indicate that the Ahr catchment was only partly hit by the most extreme precipitation cells (Fig. 2). The 24 h
precipitation totals inferred from the radar-based RADOLAN data product corroborate this observation (Mohr et
al., 2023). The northwestern part of the catchment received the highest precipitation, but a large part of the
extreme rainfall fell outside the catchment (Fig. 2). Comparison of the areas of the most intense precipitation

(Fig. 2) with the orography (Fig. 1) reveals that these areas are not necessarily associated with high elevations in
the Eifel mountains, but rather aligned with the major trajectory of the moisture transport. This suggests that the
position of the trough, which controls the moisture transport, strongly influences the location of the precipitation
footprint.

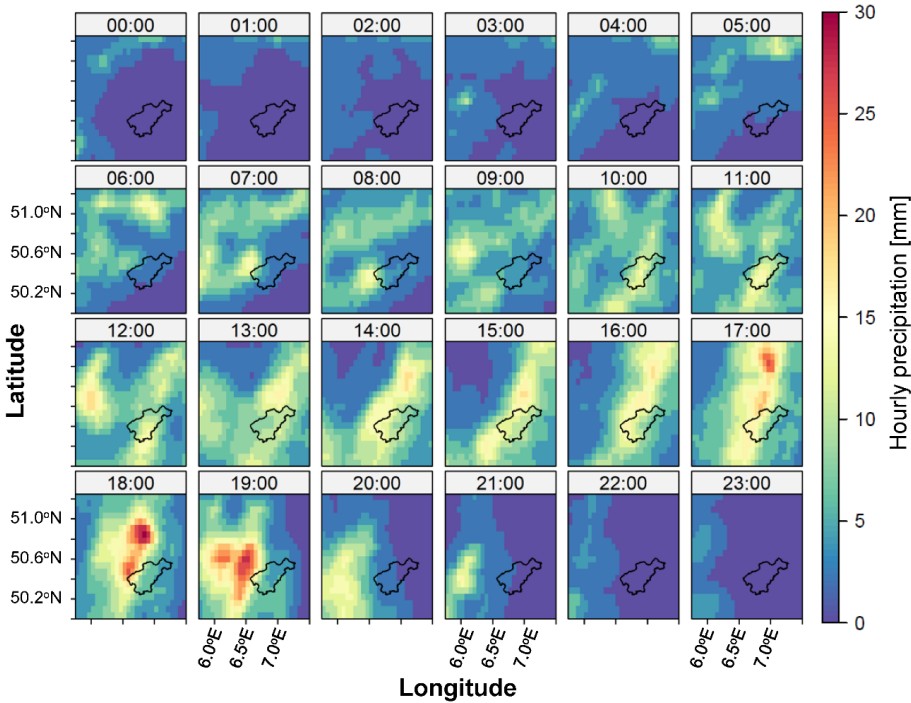

**Figure 2.** Hourly precipitation footprints from the disaggregated E-OBS dataset in the Eifel region around the
Ahr catchment for 14 July 2021, between 00:00 and 23:00 UTC.

We develop spatial counterfactuals by shifting the entire precipitation field to the west and east, mimicking
alternative positions of the low-pressure system. Additionally, we explore a few scenarios by shifting the





precipitation footprint a few kilometers eastward and then southwards (in a few steps). These counterfactuals not
only mimic an alternative position of the trough, but also alternative scenarios of peak precipitation footprints
along the major trajectory of moisture transport. We shift the precipitation field in steps of 0.0625°,
corresponding to about 4.5 km at the latitude of the Ahr catchment (50° N). We explore precipitation shifts with
3 steps westward (W1, W2, W3) and 13 steps eastward (E1 to E13). Furthermore, after shifting the precipitation
field 5 steps eastward, we investigate a range of steps along the north-south axis (E5N1 – E5N4, E5S1 – E5S5).
The latter counterfactuals explore the effect of placing the most intensive precipitation between 18:00 and 20:00
UTC of 14 July 2021 centrally over the catchment (Fig. 2). In total, 25 scenarios are examined in addition to the
reference scenario without shift (S0). The shifts are consistently applied to all hourly precipitation totals for the
period of 5 days around the event date from 12 July 00:00 to 18 July 2021 23:00 UTC.

**3.2 Hydrological model**

The hydrological response of the Ahr catchment to different spatial counterfactuals is investigated with the grid-
based mesoscale Hydrologic Model mHM (Samaniego et al., 2010, Kumar et al., 2013, Samaniego et al., 2019).
mHM is set up for the Ahr catchment with 3 x 3 km grid resolution and an hourly timestep. mHM is driven by E-
OBS precipitation fields disaggregated from daily to hourly values as described in Sect. 3.1. Hourly air
temperature $T_j (j = 1, 2, \cdots 24)$ is disaggregated from E-OBS daily maximum ($T_{max}$) and minimum ($T_{min}$) air
temperature using a cosine function (Förster et al., 2016):

$$T_j = T_{min} + \frac{T_{max} - T_{min}}{2}\left(1 + cos\frac{\pi(j+a)}{12}\right), \tag{1}$$

where $a$ controls the time of daily maximum temperature within a day. The value of $a$ is calibrated based on
observed hourly air temperature data and is set to 8 from January to April and 9 for other months for the study
area.

The land surface characteristics required by mHM include a digital elevation model acquired from the Federal
Agency for Cartography and Geodesy (BKG) and a digitized soil map from the Federal Institute for Geosciences
and Natural Resources (BGR). Based on these maps, we extract information on soil texture properties, hydraulic
conductivities, and topographic properties (such as slope, aspect, flow direction and flow accumulation). Land
cover information is derived from CORINE land cover scenes of the years 2000, 2006, 2012, and 2018
(European Environmental Agency, EEA). mHM utilizes the Multiscale Parameter Regionalization (MPR)
(Samaniego et al., 2010) technique for consistent parameterization across space resulting in a consistent
parameter set for the entire model domain. The hydrological model is calibrated in a multi-site framework based
on the Dynamically Dimensioned Search (DDS) algorithm (Tolson and Shoemaker, 2007) using 6 years (2016-
2021) of hourly discharge time series at the gauges Müsch and Altenahr. Unfortunately, due to gauge failure, no
instrumental records are available for the July 2021 event. We thus use the reconstructed flow hydrographs for
model calibration considering high-water marks, inundation extents and downstream gauge records as described
by the Environment Agency of Rhineland-Palatinate (Berker et al., 2022). We use the modified weighted Nash-
Sutcliffe Efficiency (wNSE) as the objective function which puts a stronger weight on predicting high peak
flows (Hundecha and Merz, 2012):


$$wNSE = 1 - \frac{\sum_{i=1}^{N} Q_o(t_i)\left(Q_s(t_i) - Q_o(t_i)\right)^2}{\sum_{i=1}^{N} Q_o(t_i)\left(Q_o(t_i) - \overline{Q_o}\right)^2}$$ (2)

where $Q_o(t_i)$ and $Q_s(t_i)$ are the observed and simulated discharges at time step $t_i$; $\overline{Q_o}$ is the mean observed discharge over the period of N time steps. Additionally, the Kling-Gupta Efficiency, the Nash-Sutcliffe Efficiency and the percentage difference between the maximum simulated and reference discharge (i.e., observed or reconstructed) are used to characterize the model performance. Calibration and validation of

hydrological models are typically performed using a split-sample approach (Klemeš, 1986). Since the July 2021 flood was an exceptional event, it should be included in both the calibration and validation. Given the presence of exceptional runoff conditions with widespread overland flow and very high runoff coefficients, it would be naïve to expect the model to capture such an event with parameters calibrated without this event. Finally, the validation should also include the 2021 flood, since this is the target event for the developed model. Hence, we

need to adopt a different calibration and validation approach than a split-sample test. Here we use a spatial validation approach: We calibrate the model at the gauges Müsch and Altenahr and validate the model at 5 gauges (Kreuzberg, Denn, Kirmutscheid, Niederadenau, and Bad Bodendorf; see Fig. 1), which are not used for calibration.

For the analysis of spatial counterfactuals, mHM is driven by different precipitation scenarios, whereas the

temperature field is kept constant in space. Each mHM model run uses a warm-up period of 5 years (2016-2020) prior to the July 2021 flood. The shifted precipitation field for the period of 5 days is inserted into the time series. Hence, spatial counterfactuals are applied to the factual simulated antecedent catchment conditions. Nevertheless, small deviations from the real antecedent soil moisture state occurs during the 5 days, where shifting is applied. They can be considered relatively small since no strong rainfall events occurred in this period.

**3.3 Hydrodynamic model**

We use the raster-based two-dimensional hydrodynamic model RIM2D which solves a simplified shallow water equation. This so-called local inertia approximation disregards the convective acceleration term of the momentum equation and can be solved efficiently in the explicit manner (Bates et al., 2010). The inertia formulation has been previously evaluated in a number of synthetic tests (Bates et al., 2010) and real-case

applications (e.g. Neal et al., 2011). Numerical instability occurring at super-critical flows can be efficiently tackled by introducing numerical diffusion, as proposed by de Almeida et al. (2012). RIM2D is coded in CUDA Fortran and parallelized for NVIDIA Graphical Processor Units (GPUs). This efficient parallelization enabled long-term continuous simulations for flood risk assessments (Falter et al., 2015, Sairam et al., 2021) and paved the avenue for operational flood inundation and impact forecasting (Apel et al. 2022). In the latter study, Apel et

al. (2022) set up the RIM2D model in a hindcast mode for the downstream part of the Ahr valley, from the gauge Altenahr down to the confluence with the Rhine River. This setup forms the basis for the analysis presented in this study and was further extended upstream for the domain Müsch-Altenahr (Fig. 1).

RIM2D runs at the spatial resolution of 5 x 5 m. The topography is represented by the respective digital elevation model (DEM5), aggregated from the 1 x 1 m DEM of the federal state of Rhineland-Palatinate. The

river channel bathymetry is poorly represented in the DEM5, i.e. the bankfull depth is underestimated. The mean long-term water depth along the Ahr river is between 0.4 m at gauge Müsch and 0.85 m at gauge Bad Bodendorf.



With water depths of up to 10 m during the July 2021 flood, Apel et al. (2022) showed that even the 10 m resolution is acceptable for simulating the inundation of this event. For the reach Altenahr – Sinzig, RIM2D exhibited a very high critical success index (CSI, Aronica et al., 2002) of 0.845 when run for event re-analysis

driven by the reconstructed water depth hydrograph (Apel et al., 2022). The comparison of simulated and reported water depths also showed very good agreement: Based on 75 high water marks recorded at buildings in the inundated areas in the aftermath of the flood, the bias between reported and simulated water depth was -0.39 m and the root mean squared error (RMSE) was 0.66 m.

The model domain used in Apel et al. (2022) is extended here to the gauge Müsch encompassing the entire reach

Müsch – Sinzig (Fig. 1). The roughness parameterization of the RIM2D model is carried out based on the Mundialis[1] land cover mapping for Germany derived from Sentinel-2 data. . The land use map is reclassified into 7 classes. 12 different sets of Manning's values for these 7 land use classes (Table S1) are tested to find the best fit to the reconstructed flow hydrograph at gauge Altenahr.

Buildings are extracted from the OpenStreetMap (OSM) building layer, rasterized to the resolution of the DEM

and overlaid with the topography. Hence, buildings are treated as impermeable obstacles in the hydrodynamic simulation, i.e., water flow is simulated around the buildings.

The upstream boundary condition for RIM2D is given by the hourly water depth hydrograph at gauge Müsch. The water depth is estimated from discharge time series simulated by mHM using the official gauge rating curve. The lateral inflow of the gauged tributaries Kirmutscheid, Niederadenau, Denn and Kreuzberg is added by

assigning the water depth hydrographs derived from the gauge rating curves from mHM discharges at the tributary mouths. For the 9 ungauged tributaries (Figure 1) synthetic rating curves are derived to convert mHM discharge to water levels using Manning's equation. The wetted perimeter, slope and gauge datum (= bed elevation) are derived from the DEM5 and a roughness value of n = 0.03 is assumed. The resulting water level hydrographs are provided as lateral boundaries to RIM2D. At the downstream boundary, the normal depth

condition is assumed, i.e., the water level gradient is the same across the domain boundary as for the previous two cells at the boundary. The two RIM2D simulation domains are initialized with steady-state conditions corresponding to the discharge just before the flood wave. For gauge Müsch, this was Q = 9 $m^3$ $s^{-1}$ (0.8 m water depth), which corresponds to the discharge recorded on 14 July 10:00. For gauge Altenahr, the discharge from the initial phase of the rising limb of the flood event with Q = 130 $m^3$ $s^{-1}$ (2.68 m water depth) at July 14th 17:00

is used. The simulations are continued until the steady state is established and the river channel represented by the DEM is filled.

### 3.4 Simulation experiments and evaluation procedure

All 25 spatial counterfactuals are simulated with mHM and RIM2D in addition to the reference scenario (S0). The mHM results in terms of peak flow and event volume are compared for all 7 gauges in the Ahr catchment.

From all simulations with RIM2D, we select two counterfactuals with the highest and the lowest water level at the gauge Altenahr and compare them to the reference scenario. To evaluate the changes in the resulting

---

[1] https://www.mundialis.de/en/germany-2020-land-cover-based-on-sentinel-2-data/




inundation areas, maximum and mean water depths, we select 11 focus areas (Figure 3) and compute the respective changes.

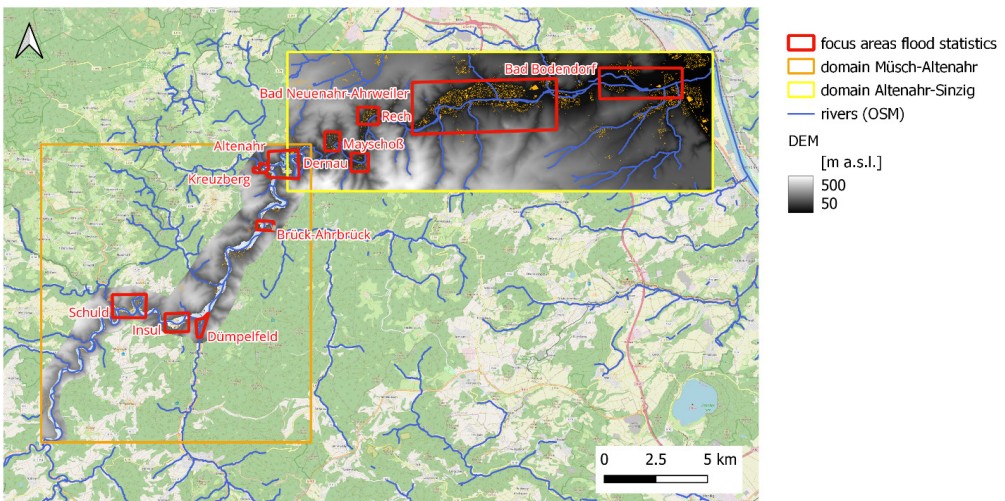

**Figure 3.** Overview of 11 focus areas used for evaluation of spatial counterfactuals with regards to inundation impact. These areas were particularly hit by the July 2021 flood.

## 4. Results

### 4.1 Antecedent and event precipitation in the counterfactual scenarios

For all scenarios, we compute the mean areal precipitation of the catchment gauged at Altenahr. We aggregate the event precipitation into 1, 3, 6, 12 and 24 h totals and find the maximum totals during the event (Table 1). Additionally, we aggregate the 2-day precipitation prior to 14 July 2021 to determine antecedent precipitation. For all counterfactuals with eastward shifting, we detect higher antecedent and event precipitation for all aggregation time windows compared to the S0 scenario. The westward shifts result in a gradual decrease of all

precipitation indicators (Table 1). The scenarios E3 and E4 stand out as they exhibit the highest precipitation values at 6 h,12 h and 24 h aggregation steps.

**Table 1.** Total maximum areal precipitation for 1, 3, 12, 24 h durations and antecedent 2-day precipitation prior to 14 July 2021 0:00 UTC for all spatial counterfactuals and the reference scenario for the Altenahr catchment. Maximum values are in bold.

| Scenarios | 1h [mm] | 3h [mm] | 6h [mm] | 12h [mm] | 24h [mm] | Antecedent precipitation [mm] |
|---|---|---|---|---|---|---|
| W3 | 8.0 | 23.6 | 44.7 | 70.8 | 72.2 | 15.3 |
| W2 | 8.7 | 25.3 | 49.1 | 78.6 | 80.8 | 15.9 |
| W1 | 9.3 | 27.7 | 52.8 | 86.1 | 89.6 | 16.4 |
| S0 | 10.3 | 29.8 | 56.7 | 92.9 | 97.9 | 16.8 |
| E1 | 11.0 | 31.5 | 60.7 | 98.0 | 105.0 | 17.1 |



| | | | | | | |
|---|---|---|---|---|---|---|
| **E2** | 11.6 | 33.8 | 64.6 | 101.2 | 110.2 | 17.5 |
| **E3** | 12.8 | 36.9 | 66.9 | **102.2** | 113.3 | 17.8 |
| **E4** | 14.5 | 39.0 | **67.6** | 101.0 | **114.3** | 18.2 |
| **E5** | 15.7 | 39.8 | 66.2 | 97.8 | 113.4 | 18.8 |
| **E6** | 16.6 | 39.6 | 63.1 | 93.0 | 111.1 | 19.5 |
| **E7** | 17.2 | 38.4 | 58.7 | 87.6 | 108.7 | 20.8 |
| **E8** | 17.5 | 36.6 | 54.0 | 83.5 | 107.0 | 22.5 |
| **E9** | 17.2 | 35.4 | 50.8 | 79.8 | 105.2 | 23.6 |
| **E10** | 16.4 | 34.1 | 48.8 | 76.8 | 103.6 | 24.1 |
| **E11** | 15.2 | 32.3 | 46.2 | 73.6 | 102.0 | 23.8 |
| **E12** | 13.8 | 31.3 | 43.5 | 70.7 | 101.6 | 23.5 |
| **E13** | 12.6 | 30.5 | 41.4 | 68.5 | 102.8 | 23.1 |
| **E5N4** | 11.3 | 29.0 | 55.5 | 94.0 | 101.5 | 31.2 |
| **E5N3** | 11.5 | 31.6 | 59.4 | 96.5 | 106.1 | 28.2 |
| **E5N2** | 12.8 | 34.1 | 62.3 | 98.0 | 109.3 | 24.2 |
| **E5N1** | 14.3 | 37.0 | 64.7 | 98.4 | 111.9 | 20.7 |
| **E5S1** | 17.2 | 42.3 | 66.5 | 96.6 | 113.5 | 20.0 |
| **E5S2** | **18.0** | 43.9 | 65.2 | 94.9 | 112.3 | 23.3 |
| **E5S3** | 17.9 | **44.2** | 62.5 | 93.1 | 109.6 | 27.9 |
| **E5S4** | 16.7 | 43.0 | 58.0 | 89.2 | 105.1 | 31.6 |
| **E5S5** | 15.9 | 40.5 | 52.4 | 83.2 | 99.1 | **33.8** |


### 4.2 Calibration and validation of the hydrological model

The calibration results for the gauges Altenahr and Müsch over the period from 2016 to 2021 deliver wNSE values of 0.98 and 0.97, respectively. The validation performance at the other 5 gauges ranges with wNSE values between 0.35 at gauge Kreuzberg to 0.94 at gauge Bad Bodendorf. Figure 4 illustrates the model performance at

the Altenahr gauge, including the hydrographs of the two largest floods within 2016-2021. The simulated streamflow shows a good agreement with the observations, especially for the high values including 2021 flood event. The peak difference with the reconstructed data is only 7.8%. Compared to the high wNSE value of 0.82 across all gauges, the average KGE and NSE values are relatively low at 0.59 and 0.57, respectively. This poorer performance mainly results from the overestimation of the low flow by the model.

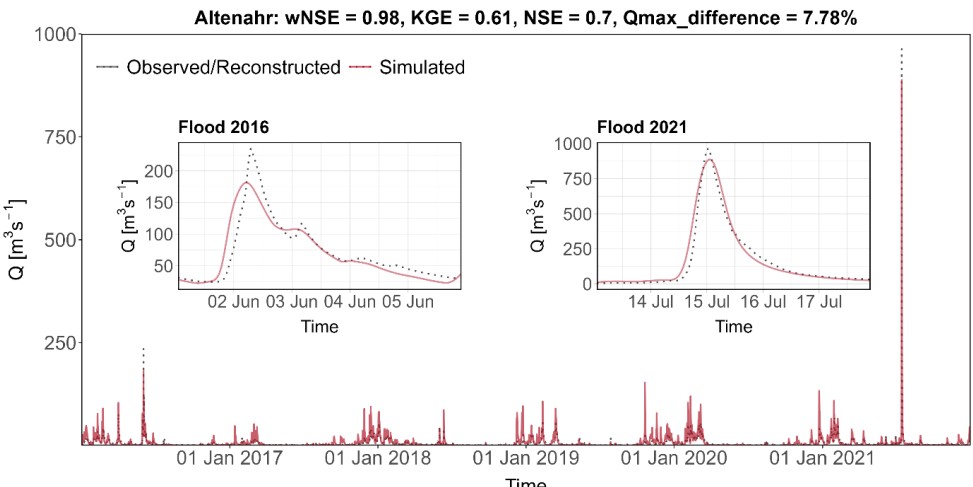

**Figure 4.** Simulated and observed/reconstructed discharge (Q) timeseries for the gauge Altenahr during 2016-2021. Two insets display the hydrographs for the two largest recorded flood events in June 2016 and July 2021.

**4.3 Discharge in the counterfactual scenarios**

All spatial counterfactuals are evaluated in terms of changes in flood peak and flood event volume with respect to the reference scenario (S0) at 7 gauges (Fig. 5). The flood volume is computed for the event duration from 13 July 00:00 to 18 July 23:00 UTC. For comparison, also the reconstructed event is plotted in the volume/peak change diagrams. For every sub-catchment, we find scenarios that result in both a larger peak flow and a larger event volume compared to the reference scenario (S0). The maximum changes are simulated at gauge Denn reaching nearly 160 % increase in peak flow and around 90 % increase in event volume for the E7/E8 counterfactuals. Besides Denn, other sub-catchments located in the south-eastern part of the Ahr catchment, including Kirmutscheid and Niederadenau, exhibit a strong reaction with more than 75 % and 100 % change in peak flow, respectively. In these two tributaries, the counterfactuals with eastward and southward shifts caused the highest peaks (E5S2/E5S3). The least sensitive sub-catchment is Müsch with the maximum peak and volume increase of about 20%. With a westward precipitation shift, the peak flow decreases up to 30% (W3). The precipitation footprint in the reference scenario (S0) shows already intensive rainfall in the north-western part of the Ahr catchment (Fig. 2) and represents one of the worst scenarios for the Müsch sub-catchment when compared to the shifted patterns. The westward shifts (W1 – W3) result in gradual and nearly proportional reduction of peak and volume in all sub-catchments. The gauge Denn shows here the most sensitive response with the reduction of up to 60 % (Fig. 5).

In several small sub-catchments, i.e., Denn, Niederadenau, Kirmutscheid and Kreuzberg, the counterfactuals are strongly aligned along a bisecting line. The larger sub-catchments Müsch, Altenahr and Bad Bodendorf are less sensitive and show a more mixed response, i.e., the points form clouds rather than a line (Fig. 5). The linear response is observed in the tributaries, while the mixed response is simulated for gauges at the main stream. In small catchments, the stronger the overlap between the precipitation footprint and the catchment area, the stronger the response is. In larger catchments, the inflow from different tributaries is mixed and the entire reaction is dampened.


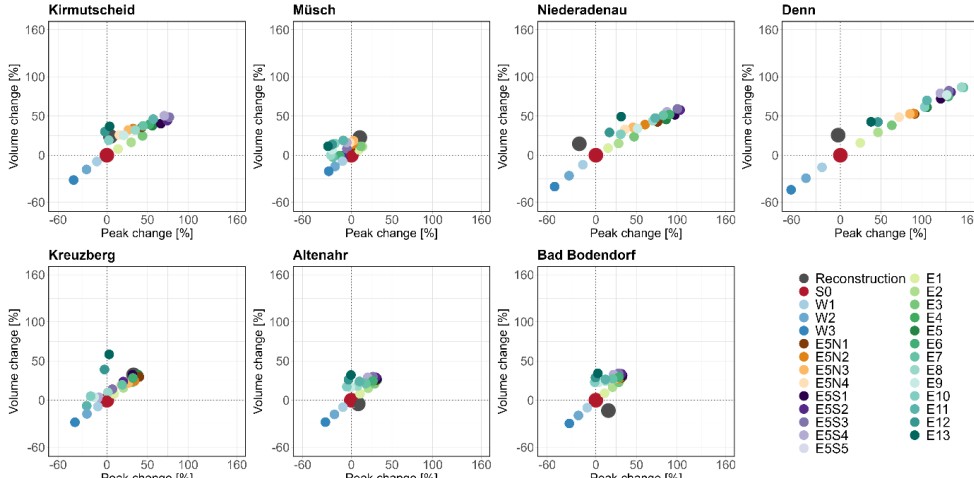

**Figure 5: Simulated changes of the flood peak and volume in the counterfactual scenarios as compared to the reference scenario (S0) at 7 gauge locations in the Ahr catchment.**

The worst counterfactuals in terms of maximum peak or volume changes are different for different sub-catchments. There is no single worst-case scenario for all tributaries which causes the worst case in the main channel. Further, there is no single clear sequence of counterfactuals with increasing order of response (increasing peak and volume) valid for all tributaries that would translate into a similar sequence of counterfactuals for gauge Altenahr. However, in general, eastward shifts of the precipitation by about 13.5 to

22.5 km (E3-E5) with some additional southward shifts (E5S1-E5S5) result in the highest peaks in almost all tributaries.

Figure 6 takes a closer look at the gauge Altenahr upstream of the major town Bad Neuenahr-Ahrweiler where widespread inundation and impacts occurred in July 2021. The worst counterfactual in terms of flood peak change is E5 (eastward shift of about 22.5 km), which results in an increase of 32 % and a corresponding flood

volume change of 26 %. Several other counterfactuals result in peak flow changes that are only a few percentage points lower (E3 (28.5 %), E5S1 (31.5 %), E5S2 (30 %)). These scenarios exhibit the highest areal precipitation in the Altenahr sub-catchment at various time scales (Table 1). Particularly, E3 scenario has the highest 6 and 12 h precipitation. Many counterfactuals with eastward shifts of only a few kilometers result in peak and volume increases of more than 10 %. The strongest volume increase of 32% (E13) is not much larger than in the highest-

peak counterfactual E5 (28%) (Fig. 6). The E13 scenario, however, delivers a small reduction of peak flow by 0.5 %.

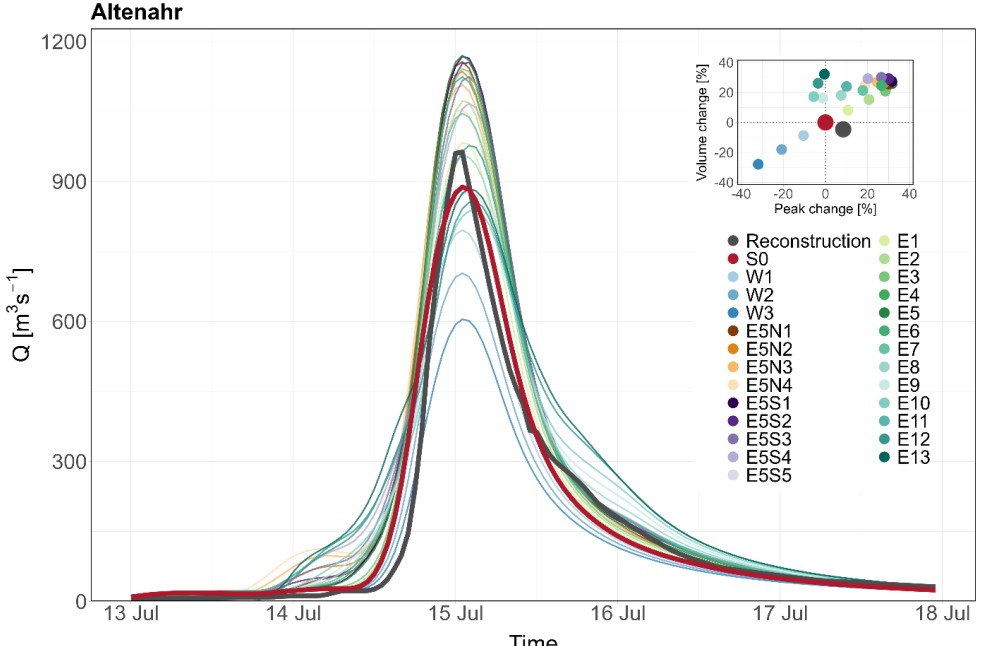

**Figure 6.** Flood hydrographs for the simulated reference scenario (S0) and spatial counterfactuals, as well as the reconstructed flood hydrograph of the July 2021 flood at the gauge Altenahr. Inset displays the peak/volume change diagram for various scenarios.

### 4.4. Calibration and validation of the hydrodynamic model

The RIM2D model is manually calibrated by testing 12 different parameter sets and comparing the resulting water level hydrographs to the hydrogaph reconstructed at the gauge Altenahr (Fig. 7). RIM2D is driven by the water level hydrograph at the gauge Müsch as an upstream boundary condition. This hydrograph is derived from the calibrated mHM flow simulations that are converted to water levels using the rating curve at gauge Müsch. Lateral inflows from the tributaries are considered as described in Section 3.3.

We select the calib4 parameter set (Table S1) as it provides the best match between the simulated and reconstructed water level hydrograph at gauge Altenahr when using mHM output as boundary conditions for the RIM2D model of the Müsch-Altenahr reach (Fig. 7). This parameter set is further used to simulate spatial counterfactual inundations. The calibrated RIM2D simulation shows higher initial water levels because of the assumed higher initial water depths and overestimates the water depths by 0.13 m, with an earlier rise of the flood limb compared to the reconstruction. On the contrary, the mHM hydrograph shows a stronger attenuation and lower peak of 0.55 m. This can in part be explained by the simple kinematic wave routing used in mHM and by uncertainty introduced by applying an extrapolated rating curve at gauge Altenahr to convert the mHM simulated flows into water levels.

For the lower reach Altenahr-Sinzig a dedicated roughness calibration was performed applying a Monte-Carlo sensitivity analysis (Khosh Bin Ghomash et al., 2024, in review). From this study, the best performing roughness


data set for simulating inundation extent, water level in the river and water depths in the floodplain was selected to simulate the counterfactual inundation in this reach.

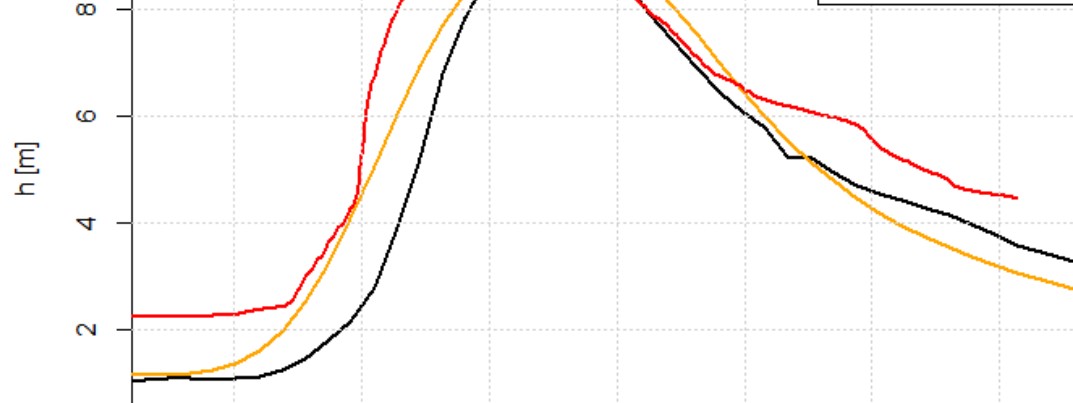

**Figure 7.** Water depth (h) hydrographs: reconstructed gauge record, simulated by RIM2D using the best roughness parameter set (calib4) and mHM S0 discharge simulation at the gauges in the reach Müsch-Altenahr, and the simulated discharge at Altenahr by mHM converted to the water depths hydrograph using the extrapolated rating curve at the gauge Altenahr.

### 4.5 Inundation in the counterfactual scenarios

We simulate inundation dynamics for all 25 spatial counterfactuals as well as for the reference scenario S0. Different counterfactuals result in different maximum water levels and inundation areas at different locations along the Ahr river. At the gauge Altenahr, for example, the maximum water level between the counterfactual scenarios simulated by RIM2D ranges between 169.20 and 172.70 m a.s.l. around 169.19 m a.s.l. corresponding to scenario S0 (Fig. 8). The span between the maximum water levels of the counterfactuals is 3.5 m, indicating how much less or more severe the flood could have been with shifting rainfall fields. In the RIM2D simulations, the highest water level is reached in scenario E3 and the lowest one in W3. These counterfactuals are selected for


further detailed analysis of changes in inundation. The range of maximum water levels converted from the mHM discharge simulations is with 4.32 m somewhat larger compared to RIM2D (Fig. 8). In the mHM ensemble of counterfactuals, the largest peak at the gauge Altenahr is obtained in E5 and the lowest in W3. As explained earlier, the different routing schemes, underlying data and conversion of discharge to water levels for mHM influence the ordering of the counterfactual scenarios, although the differences are small.

### max. water level Altenahr

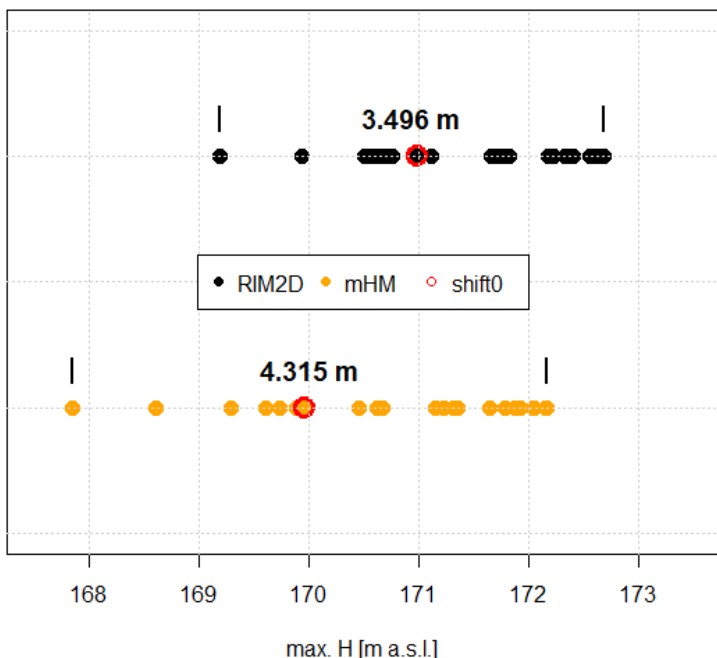

**Figure 8.** Comparison of maximum water levels at gauge Altenahr simulated with RIM2D and mHM discharge concerted by the gauge rating curve. "shift0" indicates the actual flood event with no shift of the rainfall field.

The strong discrepancy in lower water levels between RIM2D and the reconstructed scenario can be explained by assumed initial water depths in the RIM2D model, which are higher than the water levels in the Ahr before the rise of the flood hydrograph. This initial water depths were set that high to ensure a continuous flow in the 455 river channel, which bed elevation is not properly represented in the unmodified 5m DEM. Nevertheless, the peak water levels are comparable, as the effect of the high initial water depths fades out with increasing water depths. In the previous study by Apel et al. (2022), the maximum water depths in the inundated floodplains between Altenahr and the Ahr outlet were also shown to match well with field records. The higher water depths at the onset of the flood event may contribute to higher celerity in the RIM2D simulations resulting in an earlier 460 flood peak compared to the reconstructed one.

In the following step, we analyze the resulting differences in mean and maximum water depths as well as the difference in flooded area between the reference scenario (S0) and the two counterfactuals W3 and E3 (Table 2).





The fact that both W3 and E3 counterfactuals correspond to the same shift of the precipitation footprint by about 13.5 km, but in opposite directions, allows us to investigate the sensitivity of inundation characteristics to these shifts.

**Table 2.** Difference in mean and maximum simulated water depth (wd) and flooded area between the reference scenario and the spatial counterfactuals resulting in the highest (E3) and lowest (W3) maximum water depth at gauge Altenahr. *Max wd gives the 99.99 percentile of water depths to avoid potential biases by spurious maximum water depths caused by numerical instabilities and/or DEM errors.

| Focus area | W3 – S0 | | | E3 – S0 | | |
|---|---|---|---|---|---|---|
| | Mean wd [m] | Max wd [m]* | Flooded area [%] | Mean wd [m] | Max wd [m]* | Flooded area [%] |
| Schuld | -0.82 | -1.53 | -10.64 | 0.75 | 1.24 | 6.96 |
| Insul | -0.32 | -0.90 | -13.21 | 0.38 | 0.79 | 7.61 |
| Dümpelfeld | -0.36 | -2.18 | -49.59 | 0.43 | 0.98 | 19.59 |
| Brück-Ahrbrück | -0.70 | -1.27 | -15.61 | 0.79 | 1.06 | 5.54 |
| Kreuzberg | -0.78 | -1.37 | -17.42 | 0.93 | 1.45 | 9.13 |
| Altenahr | -1.50 | -2.09 | -5.68 | 1.25 | 1.75 | 4.24 |
| Mayschoß | -0.10 | -1.85 | -32.15 | 0.31 | 0.65 | 6.63 |
| Dernau | -0.59 | -0.86 | -8.61 | 0.32 | 0.46 | 1.59 |
| Rech | -1.34 | -1.64 | -5.67 | 0.52 | 0.64 | 1.89 |
| Bad Neuenahr-Ahrweiler | -0.48 | -1.07 | -11.37 | 0.19 | 0.44 | 3.53 |
| Bad Bodendorf | -0.42 | -0.64 | -8.36 | 0.15 | 0.23 | 2.48 |

The results are presented for 11 selected focus areas. For all areas, E3 leads to a consistent increase of all three inundation characteristics, whereas W3 results in a consistent decrease. In the E3 scenario, the mean and


maximum water depths are 1.25 and 1.75 m higher, respectively, than in S0 in the area of Altenahr. In the adjacent area at Kreuzberg, these numbers reach 0.93 and 1.45 m. In these topographically constricted areas,

changes in inundated areas are relatively small around 4 and 9 % (Table 2, Fig. 9).

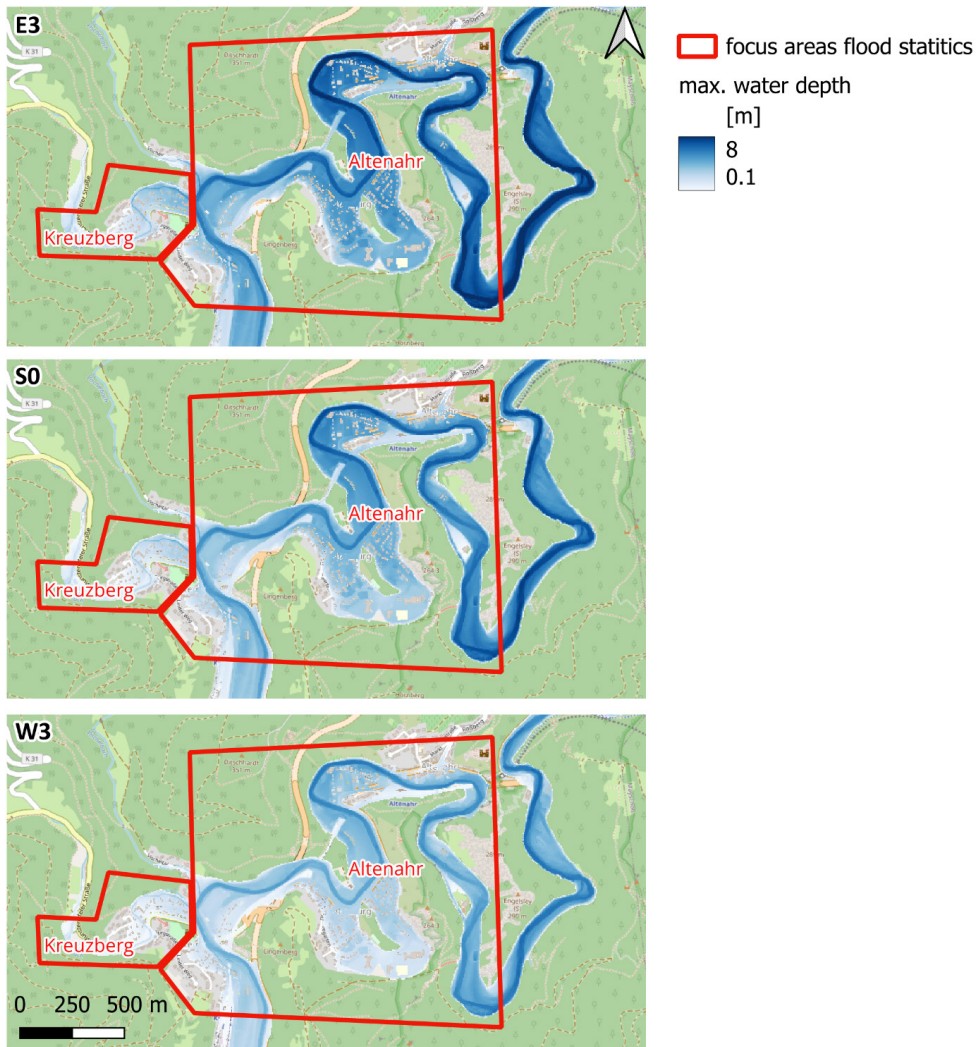

**Figure 9.** Comparison of inundated areas and maximum water depths in the focus areas of Altenahr and Kreuzberg in the E3, S0 and W3 scenarios.

The largest increase in the flooded area of nearly 20 % in E3 is detected in the village Dümpelfeld at the mouth

of the Adenauer Bach tributary (Fig. A1). Here, also the maximum water depth is higher by nearly 1 m compared to S0. This area also shows the largest inundation area decrease of nearly 50 % in the W3 scenario. The Adenauer Bach catchment has a distinct north-south orientation (Fig. 3). Hence, it becomes highly sensitive to the shifts of the precipitation footprint along the east-west axis. Three focus areas that were severely hit by the flood and experienced large damages – Schuld, Insul and Brück-Ahrbrück – also show a relatively high





sensitivity of the inundation indicators to the east-west shifting (Fig. A2, A3 and A4). Particularly at Schuld,
with narrow, deeply incised valley, the mean and maximum water depths show strong variations between E3, S0
and W3.

In Mayschoß, the eastward shift results in a comparatively modest increase in inundation area of about 7 % and
an increase of mean and maximum water depths of 0.31 m and 0.65 m, respectively, compared to S0 (Table 2).

However, W3 results in a dramatic relief for this focus area: the inundation extent is reduced by 32 %, since the
settlement area in the southern part of the village is entirely spared from flooding (Fig. 10). The sensitivity of the
inundation indicators reduces substantially downstream of Altenahr and Mayschoß in the areas of Dernau, Bad
Neuenahr-Ahrweiler and Bad Bodendorf with the exception of Rech (Table 2).

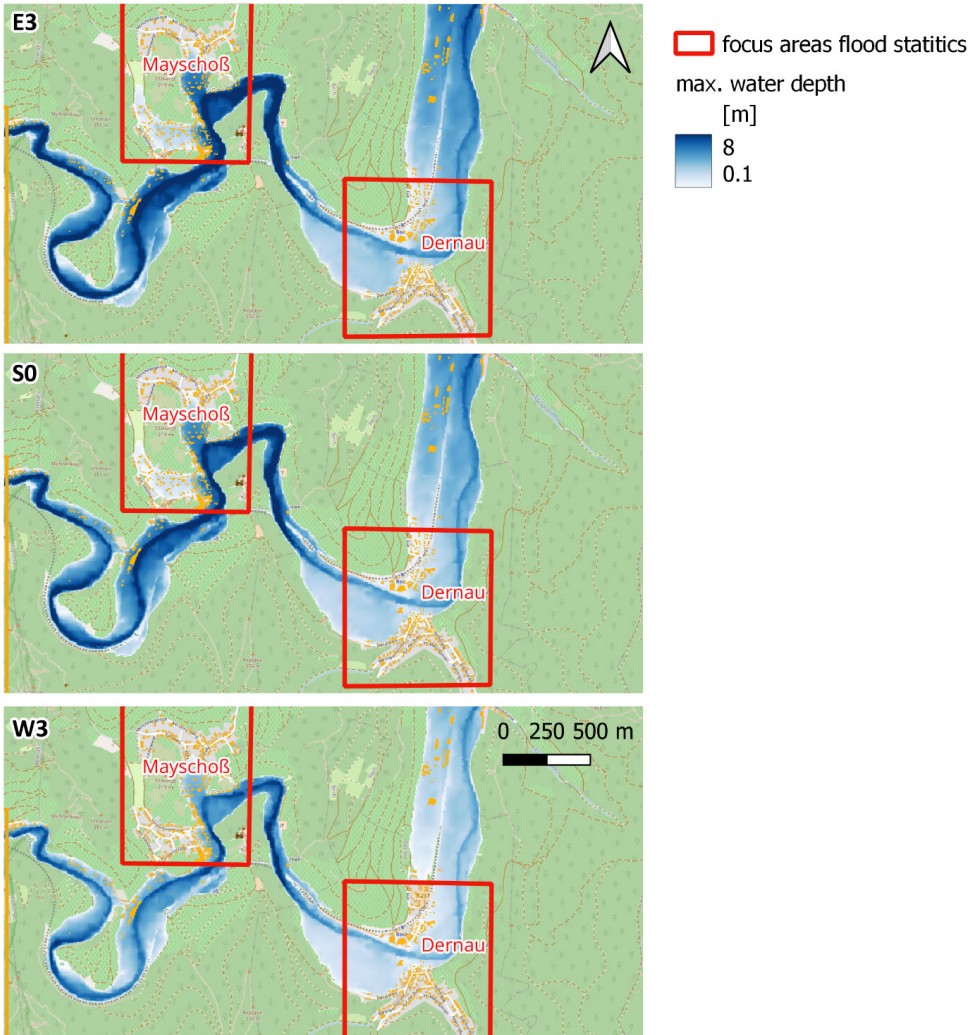

**Figure 10.** Comparison of inundated areas and maximum water depths in the focus areas of Mayschoß and
Dernau in E3, S0 and W3 scenarios.

In the downstream areas with wider floodplains at Bad Neuenahr-Ahrweiler and Bad Bodendorf the simulated changes in inundated areas and water depths are comparatively small between the analyzed counterfactuals. Mean water depths differences vary between -0.48 m and 0.19 at Bad Neuenahr-Ahrweiler and -0.42 m and 0.15

m at Bad Bodendorf. These two focus areas experienced the highest number of flood victims in the Ahr catchment. Although the increase in inundation areas and flood depths is not dramatic for E3, the relief for the comparable westward shift is stronger, but still smaller compared to other focus areas (Table 2, Fig. 11, Fig. A6). Here, most of the settlement areas remain exposed to flood waters. The lower sensitivity of the downstream areas can be expected, as they integrate the discharge from different tributaries and parts of the catchment. So, lesser

precipitation input and less severe flooding in some sub-catchments is compensated by more severe flooding of the others.

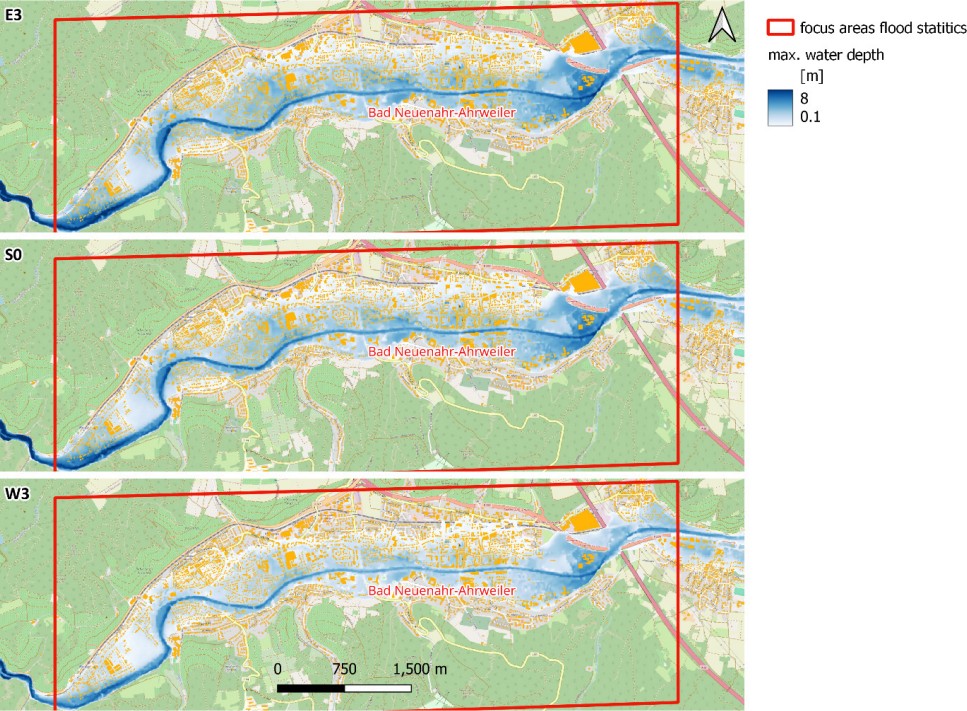

**Figure 11.** Comparison of inundated areas and maximum water depths in the focus area of Bad Neuenahr-Ahrweiler in the E3, S0 and W3 scenarios.

**4.6 Impact in counterfactual scenarios**

Table 3 shows the affected buildings in the focus areas for the W3, S0, and E3 scenarios. Affected buildings are determined by buildings, whose footprint plus a buffer of 2 m around the footprint have a mean inundation depth above 0.0 m. The buffer was used, because the footprints are rasterized to 5 m resolution, thus loosing some detail, and because the footprints are excluded from the hydraulic simulations, thus always showing inundation

depths of 0.0 m in the rasterized representation. The percentage of affected buildings in the focus areas ranges from 52.1 % at Bad Bodendorf up to 87.9 %at Rech (Table 3). These number increase to 54.1 % at Bad Bodendorf to 90.2 % at Insul in the E3 scenario. The range of the affected buildings between the scenarios





depends on the topography of the individual focus areas, but also on the different tributary discharge and consequent inundation. In the case of Dümpelfeld, the large difference in inundation is caused by the much

higher discharge of the tributary brook Adenauer Bach in E3. In this case also the fire brigade station is affected, which was not the case during the actual event (Fig. 12). The inundation depth is moderate (around 0.2 m), but this could have impaired the responsiveness of this brigade.

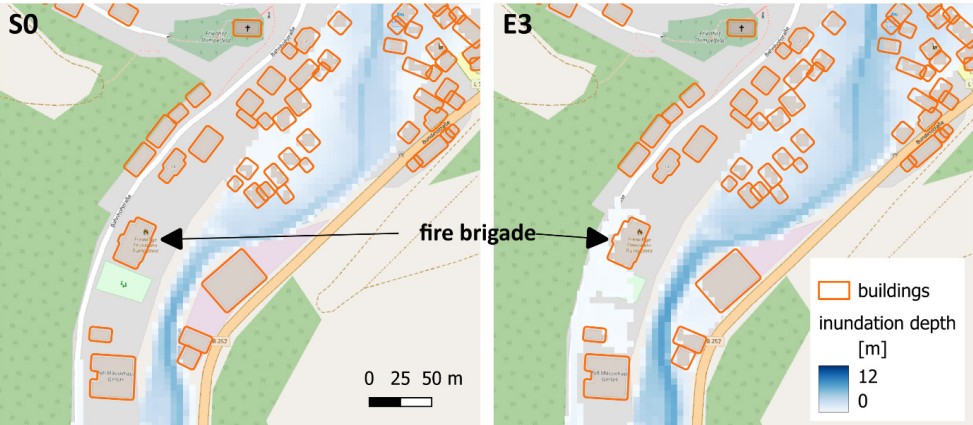

**Figure 12.** Maximum inundation depths in the scenario S0 (left) and E3 (right). Inundation around the fire brigade in Dümpelfeld is about 0.2 m in the E3 scenario. Building footprints shown are OSM footprints buffered with 2 m. Source of background image: QSM contributors.

**Table 3.** Number and percentage of affected buildings in the focus areas for the least (W3), reference (S0) and most severe (E3) counterfactual scenario.

| Focus area | Buildings | W3: affected | | S0: affected | | E3: affected | |
|---|---|---|---|---|---|---|---|
| | number | number | % of total | number | % of total | number | % of total |
| Schuld | 412 | 199 | 48.3 | 218 | 52.9 | 231 | 56.1 |
| Insul | 338 | 250 | 74 | 282 | 83.4 | 305 | 90.2 |
| Dümpelfeld | 150 | 40 | 26.7 | 91 | 60.7 | 103 | 68.7 |
| Brück-Ahrbrück | 200 | 145 | 72.5 | 163 | 81.5 | 174 | 87 |
| Kreuzberg | 138 | 86 | 62.3 | 95 | 68.8 | 103 | 74.6 |
| Altenahr | 596 | 464 | 77.9 | 485 | 81.4 | 498 | 83.6 |
| Mayschoß | 347 | 140 | 40.3 | 211 | 60.8 | 222 | 64 |
| Dernau | 300 | 105 | 35 | 157 | 52.3 | 164 | 54.7 |
| Rech | 662 | 547 | 82.6 | 582 | 87.9 | 593 | 89.6 |



| | | | | | | | |
|---|---|---|---|---|---|---|---|
| Bad Neuenahr-Ahrweiler | 6118 | 3703 | 60.5 | 4228 | 69.1 | 4365 | 71.3 |
| Bad Bodendorf | 2117 | 896 | 42.3 | 1103 | 52.1 | 1144 | 54.1 |


## 5. Discussion

The present analysis suggests that the Ahr flood catastrophe could have easily turned much worse if the trajectory of atmospheric moisture flow and the precipitation footprint were shifted just 10-20 km eastwards.
This trajectory was primarily controlled by the atmospheric circulation and the position of the low-pressure system. We could not observe a notable effect of orography on the emergence of areas of particularly heavy precipitation in the Ahr catchment. Hence, the explored shift seems to be realistic and may occur in the future. The overall low probability of such an extreme event still has to be kept in mind (Kreienkamp et al., 2021, Vorogushyn et al., 2022).

The hydrologic response of the sub-catchments to the imposed spatial counterfactuals is strikingly different. In small sub-catchments, we simulate much higher peak flows, of up to 160 % at gauge Denn, compared to the reference scenario S0. In larger sub-catchments, the response is weaker, but still notable. For example, at the gauge Altenahr, we simulate about 32 % higher peak discharge for an eastward shift of about 20 km. The same event generates more than 20 % higher flood event volume (Fig. 6). A similar maximum increase of peak flow at
gauge Altenahr by about 30 % was found by Voit and Heistermann (2024, in review), who applied a much broader range of spatial counterfactuals. Thus, this value seems to be the maximum peak flow enhancement that can be achieved by shifting such an event in space. Other modifications of the observed precipitation event are possible, such as a rotation of the precipitation footprint or changes in the overall intensity or the spatio-temporal structure. Voit and Heistermann (2022) noted that precipitation intensities can be flood-effective at different
spatial and temporal scales. Hence, by modifying the overall spatio-temporal structure of the precipitation an even stronger response cannot be ruled out. In addition, higher precipitation intensities are likely to occur in a warmer climate (Kreienkamp et al., 2021, Tradowski et al., 2023, Ludwig et al., 2023). Such changes in precipitation may be amplified by a non-linear runoff generation response, for example, when the prevailing flood generation process changes to faster overland and subsurface flow with increasing rainfall intensities
(Rogger et al., 2012, Macdonald et al. 2024). Hence, the non-linear exacerbation of peak runoff, e.g., as projected by Ludwig et al. (2023) of up to 39 % at gauge Altenahr for a +2°C warmer climate, can be even further aggravated by an unfavorable spatial counterfactual – an apocalyptic scenario beyond the imagination of decision-makers and flood-prone people.

The hydrodynamic response to the spatial counterfactuals varies between the locations along the river. We
observe different sensitivities of water depth and inundation extent at different locations to comparable shifts along the east-west axis. Small tributaries show a strong response in water depth and inundation extent at the confluence into the main channel. As can be expected from morphological characteristics, mean and maximum



water depths exhibit a strong sensitivity in constricted incised valleys. In the wider floodplains of the downstream areas at Bad Neuenahr-Ahrweiler and Bad Bodendorf, the differences in water depths between the counterfactuals are comparatively small. The strongest response of mean and maximum water depths is simulated around Altenahr. Here, shifts of the precipitation footprint in east and west direction by about 10-15 km result in changes of mean and maximum water depths by about -1.5 – +1.25 m and -2.1 – +1.75 m, respectively. In the area around Altenahr, the flow from all major tributaries is concentrated in a relatively narrow valley, before the flood wave propagates further downstream and attenuates on wider floodplains at Bad Neuenahr-Ahrweiler and Bad Bodendorf. The significant increase of maximum water depths in the worst counterfactual raises the number of affected buildings and can also potentially affect critical infrastructure such as fire brigade hubs involved into catastrophe management. Hence, spatial counterfactuals are helpful for planning and securing the operation of critical services in advance of unprecedented floods.

Here, we analyze only a limited number of spatial counterfactual scenarios, 25 in total. Precipitation fields are shifted in discrete steps in fixed directions. There is no guarantee that some other scenarios exist that might cause even higher peak flows and stronger inundation. The number of spatial counterfactuals is virtually infinite. In search for unprecedented events, Merz et al. (2024) also considered 24 spatial counterfactuals for each of the past 10 most damaging floods in Germany. They performed systematic shifts in 8 azimuthal directions and with much larger radii (20, 50 and 100 km) compared to our approach. Voit and Heistermann (2024, in review) relaxed the spatial constraints even further, by shifting the past 10 most severe precipitation events to match the centroids of more than 22,000 sub-catchments across Germany. Theoretically, even in these two cases the existence of an uncovered worst-case counterfactual cannot be ruled out.

Shifting past rainfall events to different area raises concerns about the plausibility of the occurrence of such events. This depends on the type and strength of the precipitation events as well as the moisture transport patterns and their interactions with orography. This question is largely unexplored, and it is clear that the further away a precipitation event is shifted, especially into different topographic and climatic settings, the more questionable the plausibility of its occurrence becomes. This issue can be addressed, for example, by shifting the event triggering circulation pattern in a climate model and letting the event to develop under slightly different initial conditions but constrained by the actual orography. Perturbations to past events to construct "future weather" are typically applied to explore how the event would unfold under warmer conditions (e.g. Manola et al., 2018, Ludwig et al., 2023). Similar experiments can be conceived in a stationary climate, but applying spatial counterfactuals to circulation dynamics. This approach introduces additional uncertainties through a climate model and requires additional computational effort. Also, the resulting events would not be strictly spatial transposition of the past precipitation footprints, but would unfold their own spatio-temporal dynamics. Further relaxing meteorological constraints, stochastic generation of event sets by modifying specific characteristics of the past observed events, e.g., spatial extent, total rainfall volume and peak intensity, considering their marginal statistic can be considered following the approach by Diederen and Liu (2020).

Our approach limits the number of spatial counterfactuals due to computational constraints and evaluation effort. We do not seek to find the worst possible spatial counterfactual, which may even be far from the probable maximum flood, i.e., the worst possible flood. We rather argue that even small changes in moisture flow and shifts of precipitation fields by a few kilometers may cause even more severe consequences than have been experienced. Our results should alert emergency and flood risk managers as well as the general public that the





past catastrophe was not the worst possible flood, but could have easily turned worse and thus may occur as such in the future.

The approach of spatial counterfactuals is charming from the perspective of flood risk communication as it can be easily explained and demonstrated to both flood risk professionals as well as to the general public. The approach is based on perturbing an actual past precipitation or flood event which is familiar to most people in the affected communities. Hence, people can imagine more easily the possibility of even worse catastrophe dynamics and impacts. This will hopefully increase their willingness to undertake risk reduction measures for

unprecedented events.

### 6. Conclusions

In the presented paper, we use the approach of spatial counterfactuals to explore unprecedented floods. By systematically shifting in space the footprint of the precipitation event, which caused the deadly July 2021 flood in the Ahr catchment in Germany, we simulate the resulting flood peaks, inundation areas and maximum depths

as well as exposed assets. Our findings suggest that the 2021 flood catastrophe could have been even worse if the atmospheric moisture trajectory hit the catchment only 15-25 km further east. In this case, we simulate peak flows at gauge Altenahr of about 32 % higher compared to the simulation of the actual flood. This increase in peak is associated with an increase in flood event volume of 26 %. In some small tributaries, increase in peak flows of up to 160 % is simulated in these counterfactuals. The resulting differences in inundation extents and

depths vary along the valley depending on counterfactuals and topographic properties of specific areas. For example, in a focus area around Altenahr, the mean and maximum inundation depths increase by 1.25 and 1.75 m, respectively, in the worst simulated scenario. We demonstrate that considerably more assets could have been affected by a counterfactual flood including some critical infrastructure such as a fire brigade hub. We encourage the use of spatial counterfactuals for informing flood risk professionals as well as the general public on potential

unprecedented events thus fostering better precaution and flood risk management in the years to come.

# Appendix A

**Table A1.** Parameter sets of Manning's roughness coefficients for different land use classes used for calibration of the RIM2D model.

| Land use class | calib1 | calib2 | calib3 | calib4 | calib5 | calib6 | calib7 | calib8 | calib9 | calib10 | calib11 | calib12 |
|---|---|---|---|---|---|---|---|---|---|---|---|---|
| Forest | 0.1 | 0.1 | 0.1 | 0.1 | 0.1 | 0.05 | 0.05 | 0.1 | 0.05 | 0.1 | 0.04 | 0.2 |
| Low vegetation | 0.045 | 0.045 | 0.035 | 0.035 | 0.035 | 0.035 | 0.05 | 0.05 | 0.03 | 0.025 | 0.025 | 0.05 |
| Water bodies | 0.03 | 0.03 | 0.03 | 0.03 | 0.03 | 0.03 | 0.03 | 0.03 | 0.03 | 0.03 | 0.03 | 0.03 |
| Build-up areas | 0.025 | 0.025 | 0.025 | 0.025 | 0.02 | 0.02 | 0.02 | 0.02 | 0.02 | 0.02 | 0.02 | 0.03 |





| Bare soil | 0.04 | 0.04 | 0.04 | 0.04 | 0.04 | 0.04 | 0.05 | 0.05 | 0.03 | 0.025 | 0.025 | 0.05 |
|---|---|---|---|---|---|---|---|---|---|---|---|---|
| Agricultural land | 0.04 | 0.04 | 0.04 | 0.04 | 0.04 | 0.04 | 0.05 | 0.05 | 0.03 | 0.025 | 0.025 | 0.05 |
| River channel | 0.02 | 0.018 | 0.02 | 0.018 | 0.02 | 0.02 | 0.02 | 0.02 | 0.02 | 0.018 | 0.018 | 0.025 |





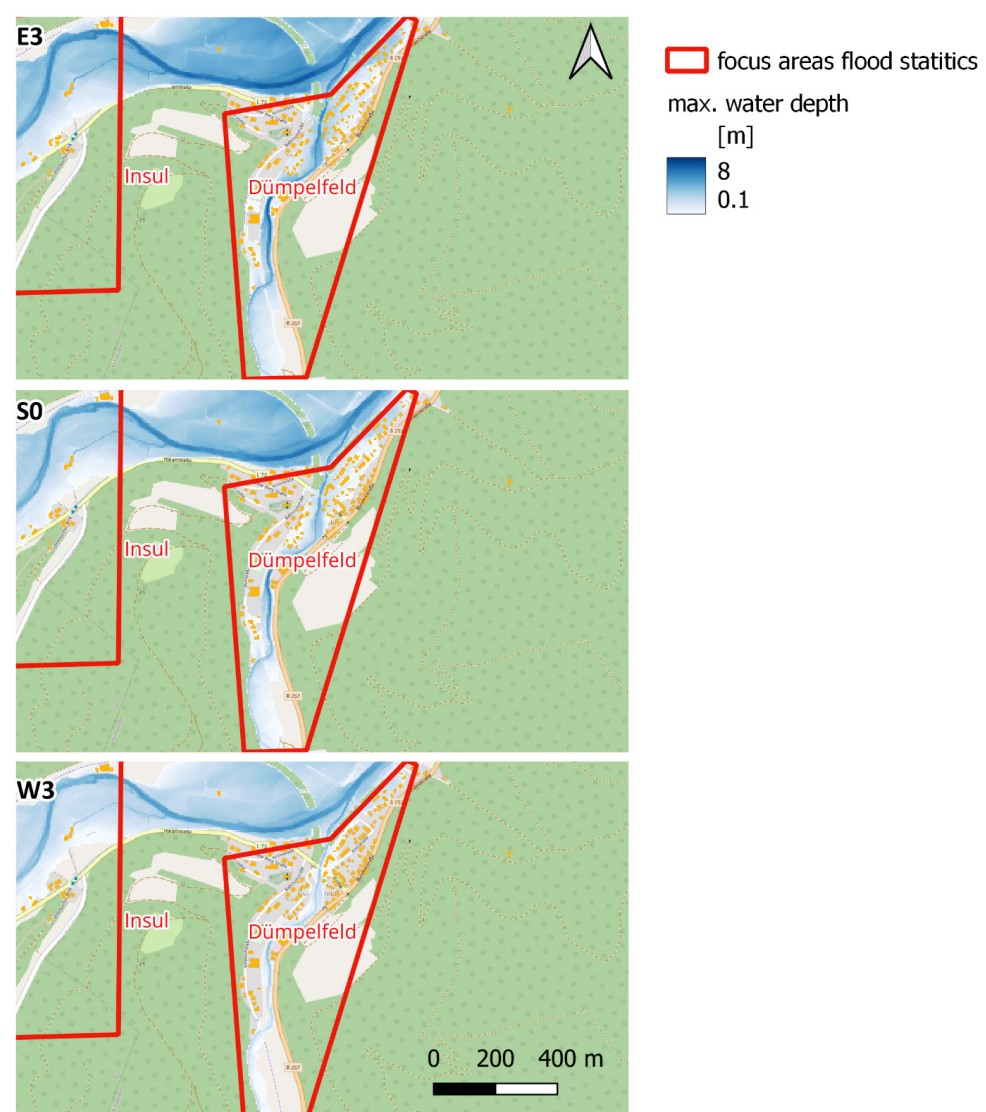

**Figure A1.** Inundation extent and maximum water depth in scenarios E3, S0, W3 in the focus area Dümpelfeld.


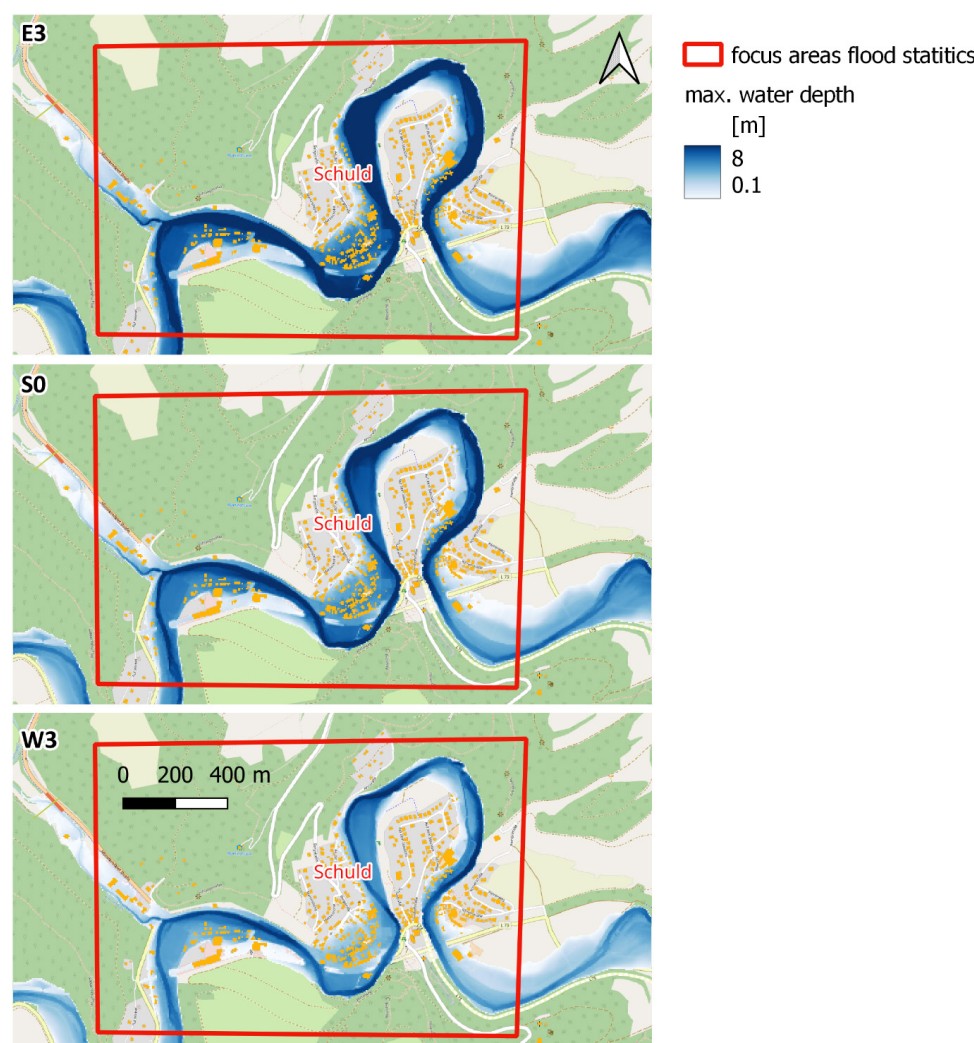

**Figure A2.** Inundation extent and maximum water depth in scenarios E3, S0, W3 in the focus area Schuld.


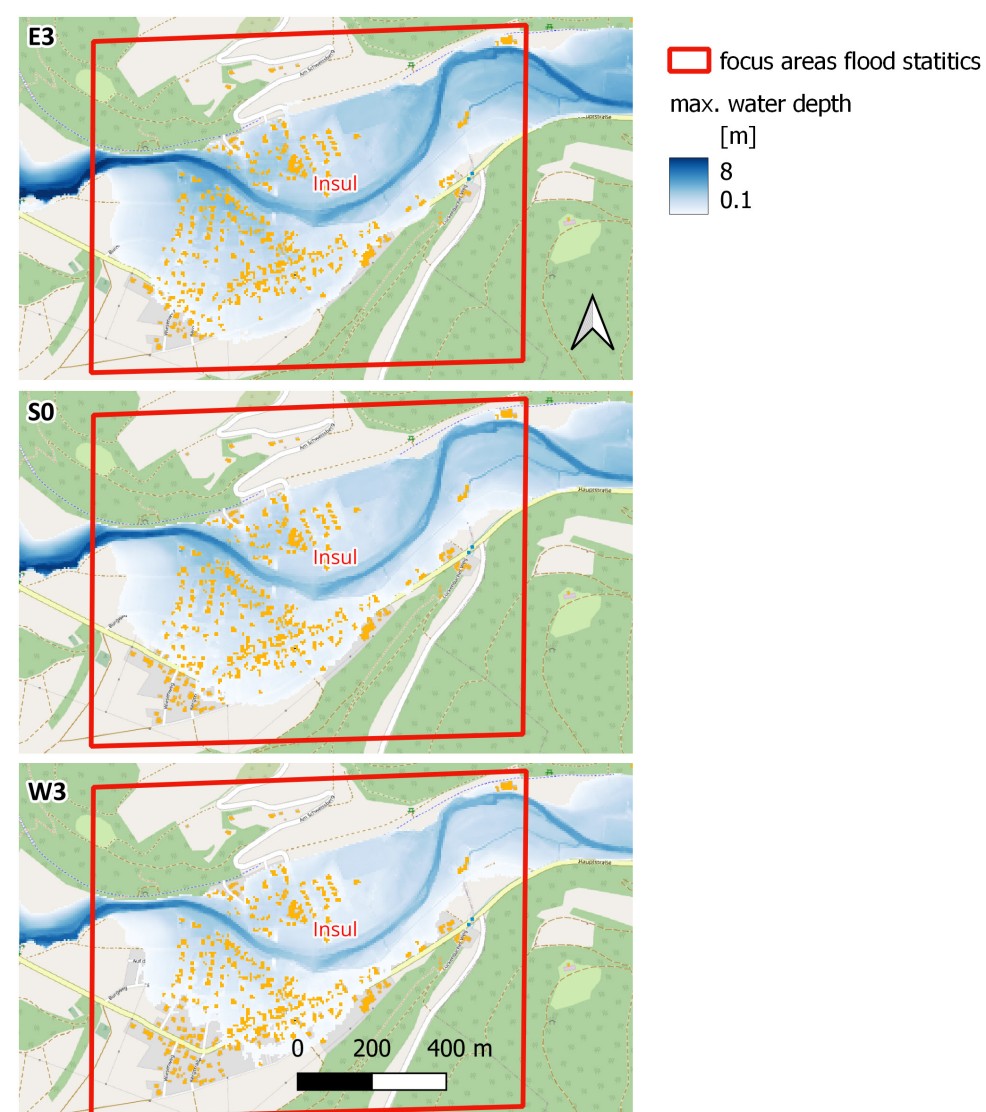

**Figure A3.** Inundation extent and maximum water depth in scenarios E3, S0, W3 in the focus area Insul.

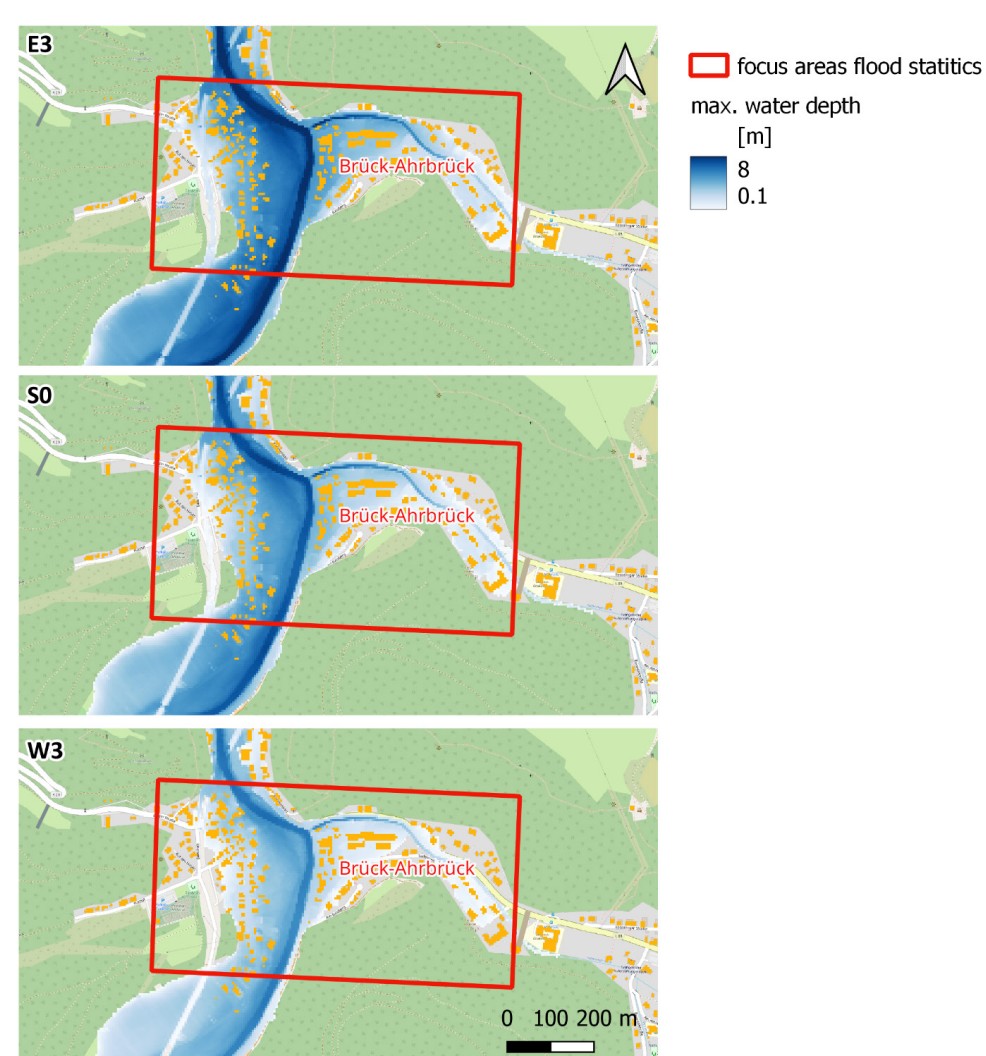


**Figure A4.** Inundation extent and maximum water depth in scenarios E3, S0, W3 in the focus area Brück-Ahrbrück.


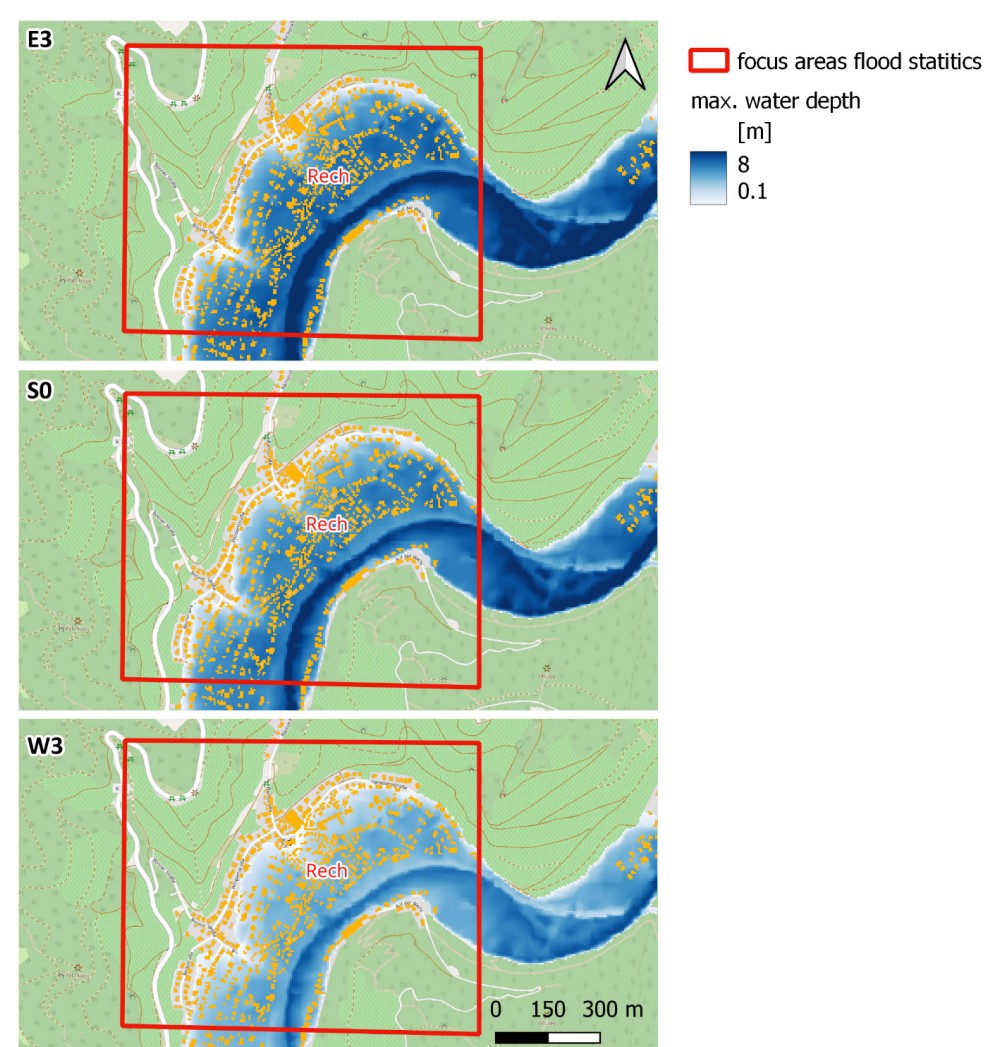


**Figure A5.** Inundation extent and maximum water depth in scenarios E3, S0, W3 in the focus area Rech.

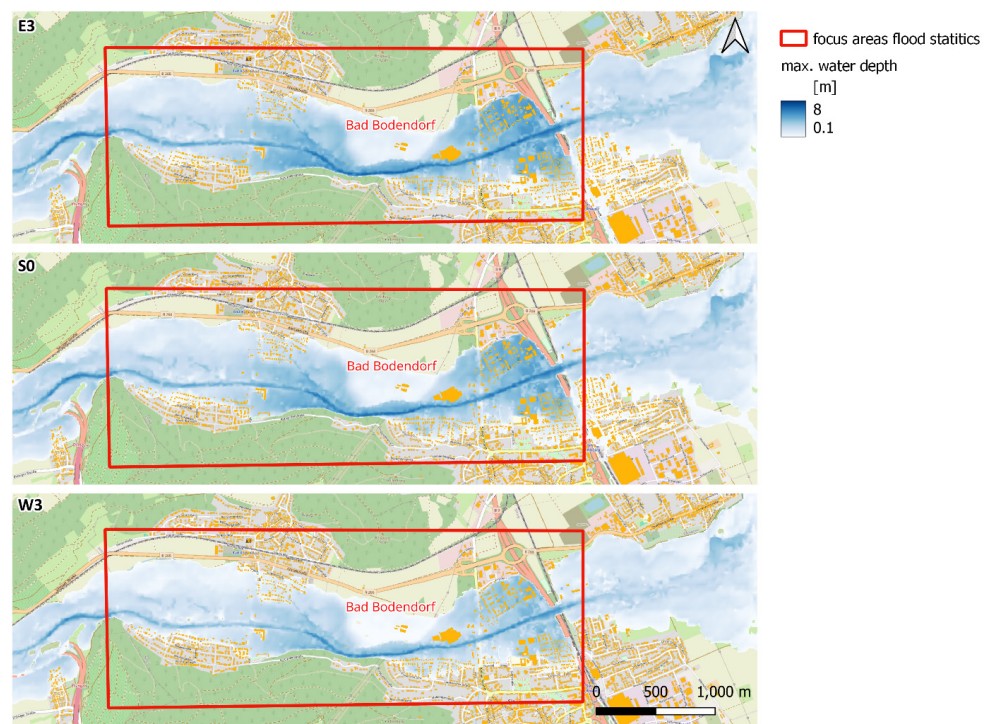


**Figure A6.** Inundation extent and maximum water depth in scenarios E3, S0, W3 in the focus area Bad Bodendorf.

*Data and code availability.* mHM model code is freely available under https://doi.org/10.5281/zenodo.8279545.

RIM2D is open source for the scientific use under the EUPL1.2 license. Access to the git repository is granted upon request. The simulations were performed with version 0.2. Simulation results of mHM and RIM2D for 25 counterfactuals and the reference scenario will be made freely available via GFZ data repository (https://dataservices.gfz-potsdam.de/portal/index.html) upon acceptance of the manuscript.

*Author contributions.* SV and BM conceived the study. DVN and XG prepared meteorological input for spatial

counterfactuals. LH and BG calibrated the mHM model, run and analyzed the hydrological simulations. HA setup the RIM2D model, run and analyzed the hydrodynamic simulations. OR, HN and LS developed and provided the mHM model version with hourly temporal resolution and the initial setup for the Ahr catchment. SV analyzed the integrated results and wrote the manuscript with contributions from all authors.

*Competing interests.* The authors declare that they have no conflict of interest.

*Acknowledgements.* This research has been supported by the German Federal Ministry for Education and Research within the KAHR project (grant number 01LR2102F). We acknowledge the E-OBS dataset from the EU-FP6 project UERRA (https://www.uerra.eu) and the Copernicus Climate Change Service, and the data



providers in the ECA&D project (https://www.ecad.eu) Digital Elevation Model (DEM) is provided by the

German Federal Agency for Cartography and Geodesy (Geobasisdaten: © GeoBasis-DE / BKG) and by the

Federal State Rhineland-Palatinate.

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
