# Peer review of "It could have been much worse: spatial counterfactuals of the July 2021 flood in the Ahr valley, Germany"

_Natural Hazards and Earth System Sciences, 2024_

## Referee Comment (RC1)

Nhess-2024-97
It could have been much worse: spatial counterfactuals of the July 2021 flood in the Ahr valley, Germany

Major Comments:

1. The authors use bi-linear interpolation to regrid the E-OBS daily precipitation data to use in the hydrological modelling. I have some concerns regarding this method of regridding. Bi-linear interpolation can smooth out extremes, which seems counterintuitive for the purpose of modelling an extreme event. Precipitation, especially at the extremes, exhibits incredibly high spatial variability, and I worry that bi-linear interpolation loses the fine-scale variations needed to accurately model the flooding. And finally, it is my understanding that bi-linear interpolation is not a conservative method, meaning that the total amount of precipitation in the coarse data may not be the same in the subsequent downscaled data. Could the authors address these concerns, add some justification for this chosen method, and perhaps provide some validation of the interpolated data, either comparing it to a more robust interpolation method or high-resolution observations? I also did not see anything in the discussion on how this choice may have affected the results.

Minor comments:

Line 130 – Why bring up snowpack and Sweden when the paper is about precipitation and Germany?

Line 195 – First use of "E-OBS" acronym – please define

Figure 7 – This figure seems to be at a lower resolution than the other figures, recommend replacing with a high res figure

Figure 8 – If this is the standard then please ignore, but my intuition says you should plot max H (m a.s.l.) along the y-axis, not the x.

Line 515 – loosing -> losing

Inundation figures (ex. Fig 9) – I like what these figures are showing, but I think they would benefit from two changes. First, I think the colorbar should be larger. Second, instead of a single column, I think they would be better represented as three columns in a single row, with scenario W3 on the left, S0 in the center, and E3 on the right. Or perhaps best would be to show the difference in the scenarios, so E3-S0 to better visualize the inundation difference between the scenarios.

---

## Author Comment (AC1)

**Reply to the comments by Reviewer #1**

Herein, the authors create a set of 25 counterfactual extreme precipitation events to simulate the catastrophic flooding seen in the Ahr Valley, Germany during July of 2021. Their use of downscaled precipitation data and hydrological modelling showed that small shifts in the trajectory of the storm systems could have resulted in even worse flooding events than what was experienced. This type of analysis shows stakeholders and policy makers how best to be prepared for natural disasters and emerging climate risks.

Overall, I found the paper to be of excellent quality. I had one major comment on the precipitation data used and a few minor comments (see attached). After those are addressed, I am confident this paper will be ready for publication.

*A: We thank the Reviewer for his/her positive comment on the manuscript and very valuable remarks that we address below.*

Nhess-2024-97
It could have been much worse: spatial counterfactuals of the July 2021 flood in the Ahr valley, Germany
Major Comments:
1.  The authors use bi-linear interpolation to regrid the E-OBS daily precipitation data to use in the hydrological modelling. I have some concerns regarding this method of regridding. Bi-linear interpolation can smooth out extremes, which seems counterintuitive for the purpose of modelling an extreme event. Precipitation, especially at the extremes, exhibits incredibly high spatial variability, and I worry that bi-linear interpolation loses the fine-scale variations needed to accurately model the flooding. And finally, it is my understanding that bi-linear interpolation is not a conservative method, meaning that the total amount of precipitation in the coarse data may not be the same in the subsequent downscaled data. Could the authors address these concerns, add some justification for this chosen method, and perhaps provide some validation of the interpolated data, either comparing it to a more robust interpolation method or high-resolution observations? I also did not see anything in the discussion on how this choice may have affected the results.

*A: Thank you for your thoughtful comment. For this study, we used a quick bilinear interpolation to adjust the resolution from 0.11° to 0.0625°, hence having minimal impact on spatial details. While bilinear interpolation may slightly smooth extremes, the close resolutions (0.11° to 0.0625°) retain key spatial features, as shown in Figure 2 and Table 1. We verified by visual check that total precipitation remained consistent before and after interpolation. Since we carry out analysis of relative changes of precipitation, discharge, volume and inundation in counterfactuals in comparison to the reference scenario, we believe the selection of this method will have little impact on the overall results and conclusions. In the revised manuscript, we shall add a short note on interpolation effects of the used approach.*

Minor comments:

Line 130 – Why bring up snowpack and Sweden when the paper is about precipitation and Germany?

*A: In our study, we apply a relatively new approach of spatial counterfactuals. This method falls into a category of methods used to construct unprecedented or worst-case scenarios. Introduction gives an overview of these methods including among others „perfect storm" approach. Whereas it is applied in other risk domains, in hydrology, we found only one mentioned example from Sweden. We believe the overview of these methods is important to properly embed the presented work in the context of research on extreme flood scenarios.*

Line 195 – First use of "E-OBS" acronym – please define

*A: Will be addressed.*

Figure 7 – This figure seems to be at a lower resolution than the other figures, recommend replacing with a high res figure

*A: This Figure will be replaced.*
Figure 8 – If this is the standard then please ignore, but my intuition says you should plot max H (m a.s.l.) along the y-axis, not the x.

*A: We shall exchange the axes.*

Line 515 – loosing -> losing

*A: Thanks. It will be corrected.*

Inundation figures (ex. Fig 9) – I like what these figures are showing, but I think they would benefit from two changes. First, I think the colorbar should be larger. Second, instead of a single column, I think they would be better represented as three columns in a single row, with scenario W3 on the left, S0 in the center, and E3 on the right. Or perhaps best would be to show the difference in the scenarios, so E3-S0 to better visualize the inundation difference between the scenarios.

*A: We agree with the reviewer that showing the difference in water depth between E3/W3 and S0 scenarios would be more informative. We will address this and adjust the legend size. Below we show an example, how e.g. new Figure 9 would look like. We think however that aligning the panels horizontally instead of vertically would lead to significantly reduced size of the images and will impair readability. Especially for Figure 11, which follows the same logic as Figure 9, this would lead to very small figure panels. We therefore prefer to keep the original alignment.*

---

## Author Response (AR1)

**Revision notes in response to the comments of the Reviewer #1**

Herein, the authors create a set of 25 counterfactual extreme precipitation events to simulate the catastrophic flooding seen in the Ahr Valley, Germany during July of 2021. Their use of downscaled precipitation data and hydrological modelling showed that small shifts in the trajectory of the storm systems could have resulted in even worse flooding events than what was experienced. This type of analysis shows stakeholders and policy makers how best to be prepared for natural disasters and emerging climate risks.

Overall, I found the paper to be of excellent quality. I had one major comment on the precipitation data used and a few minor comments (see attached). After those are addressed, I am confident this paper will be ready for publication.

*A:* We thank the Reviewer for his/her positive comment on the manuscript and very valuable remarks that we address below.

Major Comments:

1. The authors use bi-linear interpolation to regrid the E-OBS daily precipitation data to use in the hydrological modelling. I have some concerns regarding this method of regridding. Bi-linear interpolation can smooth out extremes, which seems counterintuitive for the purpose of modelling an extreme event. Precipitation, especially at the extremes, exhibits incredibly high spatial variability, and I worry that bi-linear interpolation loses the finescale variations needed to accurately model the flooding. And finally, it is my understanding that bi-linear interpolation is not a conservative method, meaning that the total amount of precipitation in the coarse data may not be the same in the subsequent downscaled data. Could the authors address these concerns, add some justification for this chosen method, and perhaps provide some validation of the interpolated data, either comparing it to a more robust interpolation method or high-resolution observations? I also did not see anything in the discussion on how this choice may have affected the results.

A: Thank you for your thoughtful comment. For this study, we used a quick bilinear interpolation to adjust the resolution from 0.11° to 0.0625°, hence having minimal impact on spatial details. While bilinear interpolation may slightly smooth extremes, the close resolutions (0.11° to 0.0625°) retain key spatial features, as shown in Figure 2 and Table 1. We verified by visual check that total precipitation remained consistent before and after interpolation. Since we carry out analysis of relative changes of precipitation, discharge, volume and inundation in counterfactuals in comparison to the reference scenario, we believe the selection of this method will have little impact on the overall results and conclusions. In the revised manuscript, we addressed this point (L201-203).

Line 130 – Why bring up snowpack and Sweden when the paper is about precipitation and Germany?

A: In our study, we apply a relatively new approach of spatial counterfactuals. This method falls into a category of methods used to construct unprecedented or worst-case scenarios. Introduction gives an overview of these methods including among others "perfect storm" approach. Whereas it is applied in other risk domains, in hydrology, we found only one mentioned example from Sweden. We believe the overview of these methods is important to properly embed the presented work in the context of research on extreme flood scenarios.

Line 195 - First use of "E-OBS" acronym - please define

*A: E-OBS is apparently not an acronym, but a proprietary name (Cornes et al., 2018). We thus provide no spelling.*

Figure 7 - This figure seems to be at a lower resolution than the other figures, recommend replacing with a high res figure

A: Done

Figure 8 – If this is the standard then please ignore, but my intuition says you should plot max H (m a.s.l.) along the y-axis, not the x.

*A:* We changed the axes showing not the water level, but water depth. Hence, it is consistent with Fig. 7 and with the notation in Fig. 8 and fits better to the discussion. Swaping the axes results in the need to rearrange the legend and the annotation. This results in a suboptimal look and we therefore decided to keep the axes as they are.

Inundation figures (ex. Fig 9) – I like what these figures are showing, but I think they would benefit from two changes. First, I think the colorbar should be larger. Second, instead of a single column, I think they would be better represented as three columns in a single row, with scenario W3 on the left, S0 in the center, and E3 on the right. Or perhaps best would be to show the difference in the scenarios, so E3-S0 to better visualize the inundation difference between the scenarios.

*A: We completely reworked Figs. 9-11 and Figures in the Appendix as proposed by the Reviewer.*

Line 515 – loosing -> losing

A: Done.

References:

Cornes, R. C., van der Schrier, G., van den Besselaar, E. J. M., and Jones, P. D.: An Ensemble Version of the E-OBS Temperature and Precipitation Data Sets. Journal of Geophysical Research: Atmospheres, 123, 9391–9409, https://doi.org/10.1029/2017JD028200, 2018.

**Revision notes in response to the comments of Michael Nones (Reviewer #2)**

Dear Authors,

I really enjoyed reading your manuscript, as it is written clearly and drives all key information in the proper way.

As you can see from my comments below, I do not see critical points in the text, and I am generally in favour of its publication after revision. I hope my comments will help you in clarifying the overall approach and what we can learn from this example.

*A*: We thank Michael Nones for his positive review of the manuscript and very valuable remarks that we address below.

**General comments**

I suggest adding some more comments on the numerical model in the Abstract (which one? calibrated/validated how?), as results depend on it.

**A: A short note is added in the abstract.**

Ludwig et al. (2023) pointed out a significant role of sediment in the 2021 event. As you used a hydrodynamics model assuming clear water, could you please comment on potential uncertainties connected with river/valley morphodynamics? What about considering other floating materials constituting the accumulated debris?

*A*:Indeed, large amounts of sediment has been displaced and transported during the flood. Also e.g., extensive field survey of Dietze et al. (2022) provide some interesting insights. Still, we believe the Ahr flood cannot be regarded as non-Newtonian mudflow. Based on field information by Dietze et al. (2022), we believe sediment scouring and deposition could have exerted some localized effects on flood water stages and depths, but overall, we expect limited effect on inundation areas and depth. In previous study with the hydrodynamic model RIM2D, Apel et al. (2022) demonstrated the sensitivity of water depths to consideration of the bankfull channel of 0.85 m depth along the entire Ahr river course for this event. Inundation depths in the floodplains were found to alter within a small range of -0.1 to 0.1 m. We expect the effect of displaced sediment to be even much smaller than adding or subtracting 0.85 m along the entire river course. Finally, and most importantly, all counterfactuals and the reference scenario have the same setup. Hence, with regards to relative changes to the reference scenario, consideration of morphodynamics is expected to be negligible. The same logic basically applies to the floating debris. Dietze et al. (2022) and Ludwig et al. (2023) provide an extensive discussion on morphodynamics during the 2021 flood. Since, our work is not directly related to this issue and we expect it not to significantly alter the results, we would omit this discussion.

Figure 4 shows that the model tends to anticipate the observed/reconstructed flood wave. Does this affect your results? Could you please comment a bit more on this, while presenting the results (Sec. 4.2)

A: The slight shift in phase for the simulation of the 2016 event shown in Figure 4 does not influence our results and conclusions since we do not compare event timing or duration in our analysis. Our calibrated model captures the peak and duration of the 2021 flood well. Therefore, we use this same calibrated model simulating all counterfactual scenarios of the 2021 flood to ensure consistency and comparability.

Secs. 4.5 & 4.6: As the results are influenced by simplifications in the modelling scheme and uncertainties in the input data, I suggest providing not only final exact results, but also discussing more in detail uncertainties, providing their estimate

A: We added the following text in the Discussion section "The use of the hydrologic and hydrodynamic models for the analysis of spatial counterfactuals is associated with uncertainties in model structure and input data and parameterization. We, however, analyze relative differences in flood characteristics between various scenarios. Hence, the effect of uncertainty is expected to be fairly limited on the final results and conclusions and we do not explicitly consider these uncertainties in the presented analysis. Interestingly, Voit and Heistermann (2024) found a very similar maximum relative increase of flood peak for the Ahr flood using a different hydrological model and completely different approach to construction of counterfactuals. This confirms our expectation. In fact, we can view the analysis of spatial counterfactuals as exploration of aleatory uncertainty (natural variability) in spatial precipitation footprints."

In the Discussion, you pointed out that, out of infinite spatial counterfactual scenarios, you used only 25. Could you please provide more details on how you selected them, also expanding the Introduction by adding some more comments on the study rationale?

*A:* We added the following text to Introduction (L166-168): This analysis is expected to raise awareness for extreme events exceeding any previous experience among flood managers as well as potentially affected population and help better prepare for such scenarios and reduce risk of death toll and economic damage. We also extended the Discussion (L600-605).

From your results, one can argue that flood mapping should be done by considering multiple (potentially infinite) scenarios. Do you think that flood risk mapping and communication should be improved? How? In what direction should research go, to help communities to increase their willingness to undertake risk reduction measures for unprecedented events? This point could be discussed also in light of the very recent floods caused by storm Boris.

A: We think, our proposed methodology and the results do not directly impact already established flood hazard and risk mapping procedures as they are applied by German Federal states. Exploration of spatial counterfactuals and other unprecedented flood scenarios can however valuably extend the existing maps and help in preparing for unprecedented but plausible events, e.g., by planning new sensitive infrastructure such as hospitals, fire departments etc. outside the impact zones of such events. This is exactly the research direction, our manuscript is contributing to, which focuses on exploring a range of unprecedented extreme events and hoping to minimize negative surprise effects in case such or a similar event occurs.

**Specific comments & Technical corrections**

Figure 1: please add a world map and a map of Germany to better locate the study area for readers not familiar with the region. DEM elevation is [m asl]

A: Done

line 200: "RANDOLAN" is wrongly spelled

A: Thanks, done

line 318: please add the units of Manning's roughness n

A: Done.

line 449: you stated that "differences are small". Is it possible to have an estimate of how much small?

*A:* We compare the difference in maximum water level at gauge Altenahr between E3 scenario run with RIM2D routing and E5 scenario run with mHM and converted to water level. The resulting difference amounts 0.52 m. This difference is visible in Figure 8, and we shall mention the number in the revised manuscript and we stick to E3 scenario for further

evaluation of inundation depth differences. Since we run all counterfactuals with RIM2D routing, this difference is not decisive for the assessment of relative changes in inundation depths with respect to the reference scenario. Note is introduced (L454-457).

line 559: I suggest deleting "an apocalyptic scenario beyond the imagination of decisionmakers and flood-prone people" as it sounds a bit too personal statement and not a scientific one (even if it's true)

A: Thanks. We follow Reviewer's suggestion and delete this notion.

Figs. A1-A6: I think there is a typo in the legend, as the red box should read "... statistics"

A: We corrected the typo and reworked all Figures as suggested by the reviewer.

The work of Montanari is now published, so please update the references. Montanari, A., Merz, B., & Blöschl, G. (2024). HESS Opinions: The sword of Damocles of the impossible flood. Hydrology and Earth System Sciences, 28(12), 2603-2615. https://doi.org/10.5194/hess-28-2603-2024

Please update the reference Khosh Bin Ghomash, S., Apel, H., & Caviedes-Voullième, D. (2024). Are 2D shallow-water solvers fast enough for early flood warning? A comparative assessment on the 2021 Ahr valley flood event. Natural Hazards and Earth System Sciences, 24(8), 2857-2874. https://doi.org/10.5194/nhess-24-2857-2024

A: First reference was updated. Second reference was wrong and is exchanged.

References:

*Apel, H., Vorogushyn, S., and Merz, B.: Brief communication: Impact forecasting could substantially improve the emergency management of deadly floods: case study July 2021 floods in Germany. Natural Hazards and Earth System Sciences, 22(9), 3005–3014, https://doi.org/10.5194/nhess-22-3005-2022, 2022*

Dietze, M., Bell, R., Ozturk, U., Cook, K. L., Andermann, C., Beer, A. R., Damm, B., Lucia, A., Fauer, F. S., Nissen, K. M., Sieg, T., and Thieken, A. H. : More than heavy rain turning into fast-flowing water – a landscape perspective on the 2021 Eifel floods. Natural Hazards and Earth System Sciences, 22(6), 1845–1856, https://doi.org/10.5194/nhess-22-1845-2022, 2022.

Ludwig, P., Ehmele, F., Franca, M. J., Mohr, S., Caldas-Alvarez, A., Daniell, J. E., Ehret, U., Feldmann, H., Hundhausen, M., Knippertz, P., Küpfer, K., Kunz, M., Mühr, B., Pinto, J. G., Quinting, J., Schäfer, A. M., Seidel, F., and Wisotzky, C.: A multi-disciplinary analysis of the exceptional flood event of July 2021 in central Europe - Part 2: Historical context and relation to climate change. Natural Hazards and Earth System Sciences, 23(4), 1287–1311, https://doi.org/10.5194/nhess-23-1287-2023, 2023.

*Voit, P. and Heistermann, M.: A downward-counterfactual analysis of flash floods in Germany, Nat. Hazards Earth Syst. Sci., 24, 2147–2164, https://doi.org/10.5194/nhess-24-2147-2024, 2024.*

---

## Author Response (AR2)

Dear Editor, dear production Team,

Thank you very much for the acceptance of the manuscript.

Here final remarks to introduced manuscript changes, partly in response to the final Editor's comments, are provided:

1. All changes in the manuscript from the previous revision round have been accepted.

2. Meanwhile all papers previously cited as "in review" are published and have been now referenced in the manuscript as published papers. All citations and bibliography have been updated.

3. Bibliography has been updated using Journal Title Abbreviations by Caltech Library.

4. I declare that Figure 3 and 9-12 and all Figures in the Appendix are entirely created by the authors. Figure 3 uses Digital Elevation Model (DEM) provided by the German Federal Agency for Geodesy and Cartography (BKG) and by the Federal State Rhineland-Palatinate. This is acknowledged in the Acknowledgements section. Further, all mentioned Figures use background map, river network and building footprints from OpenStreetMap. I introduced now acknowledgement of OSM contributors in the Acknowledgements section: "Figures 1, 3, 9-12, A1-A4 use background topographic map, river network and building footprints from OpenStreetMap (© OpenStreetMap contributors).". In Figure 12, the statement "Source of background image: QSM contributors." has been removed.

5. A reference to the dataset produced in this study is now added the Data and code availability section: "Spatial counterfactual precipitation and simulation results from mHM and RIM2D for 25 counterfactuals and the reference scenario are freely available from Vorogushyn et al. (2025)." The dataset is stored in the open-access repository.

Vorogushyn, S., Han, L., Apel, H., Nguyen, V. D., Guse, B., Guan, X., Rakovec, O., Najafi, H., Samaniego, L., Merz, B.: Spatial counterfactuals of the July 2021 flood in the Ahr valley, Germany. GFZ Data Services. https://doi.org/10.5880/GFZ.RDOQ.2025.002, 2025.

For further questions, I stay at your disposal.

With kind regards,
Sergiy Vorogushyn on behalf of all co-authors.